# Non-rhythmic head-direction cells in the parahippocampal region are not constrained by attractor network dynamics

Olga Kornienko, Patrick Latuske, Mathis Bassler, Laura Kohler, Kevin Allen*

Department of Clinical Neurobiology, Medical Faculty of Heidelberg University and German Cancer Research Center, Heidelberg, Germany

**Abstract** Computational models postulate that head-direction (HD) cells are part of an attractor network integrating head turns. This network requires inputs from visual landmarks to anchor the HD signal to the external world. We investigated whether information about HD and visual landmarks is integrated in the medial entorhinal cortex and parasubiculum, resulting in neurons expressing a conjunctive code for HD and visual landmarks. We found that parahippocampal HD cells could be divided into two classes based on their theta-rhythmic activity: non-rhythmic and theta-rhythmic HD cells. Manipulations of the visual landmarks caused tuning curve alterations in most HD cells, with the largest visually driven changes observed in non-rhythmic HD cells. Importantly, the tuning modifications of non-rhythmic HD cells were often non-coherent across cells, refuting the notion that attractor-like dynamics control non-rhythmic HD cells. These findings reveal a new population of non-rhythmic HD cells whose malleable organization is controlled by visual landmarks.

DOI: https://doi.org/10.7554/eLife.35949.001

*For correspondence: allen@uni-heidelberg.de

Competing interests: The authors declare that no competing interests exist.

## Introduction

Efficient navigation in mammals depends on several classes of spatially selective neurons. The animal's sense of direction is encoded by head-direction (HD) cells that track ongoing angular movements of the head (*Taube et al., 1990a*; *Valerio and Taube, 2012*; *Butler et al., 2017*). Each HD cell fires maximally when the head of an animal points in a particular direction, with different cells having different preferred directions. HD cells are located in both subcortical and cortical areas including the dorsal tegmental nucleus, lateral mammillary nucleus, anterodorsal thalamic nucleus, nucleus reuniens, postsubiculum, retrosplenial cortex, parasubiculum, and the medial entorhinal cortex (*Taube et al., 1990a*; *Chen et al., 1994*; *Taube, 1995*; *Stackman and Taube, 1998*; *Sharp et al., 2001*; *Cacucci et al., 2004*; *Sargolini et al., 2006*; *Jankowski et al., 2014*). The HD signal is generated in subcortical areas, possibly in the dorsal tegmental and lateral mammillary nuclei, and depends on intact vestibular inputs (*Stackman and Taube, 1997*; *Muir et al., 2009*; *Yoder and Taube, 2009*; *Valerio and Taube, 2016*). This signal reaches cortical areas via the thalamocortical excitatory projections of the anterodorsal thalamic nucleus. Accordingly, lesions to the subcortical components of the HD system strongly impair HD representations in cortical areas (*Goodridge and Taube, 1997*; *Bassett et al., 2007*; *Winter et al., 2015*).

Continuous attractor network models provide a mechanistic framework explaining how the HD signal is generated (*Skaggs et al., 1995*; *Redish et al., 1996*; *Zhang, 1996*). In these models, neurons are conceptually arranged in a circle according to their preferred firing direction. The connectivity between any two neurons depends on their relative position. While neighboring cells with similar

preferred HD excite each other, distant cells tend to inhibit each other. This connectivity leads to the emergence of an activity packet which represents the moment-to-moment HD of the animal. These models predict that the differences between the preferred directions of HD cells never change because the connections within the network are immutable, and most empirical data support this prediction. For example, rotation of a distal visual landmark in an environment causes an equivalent rotation of the preferred direction of all HD cells (*Taube et al., 1990b*). When distal and proximal cues are rotated in opposite directions, HD cells in the anterior thalamus rotate coherently with one of the two sets of cues (*Yoganarasimha et al., 2006*). The firing associations of HD cells (i.e. their tendency to fire together) have also been compared across different brain states (*Tsanov et al., 2014*; *Peyrache et al., 2015*). As predicted by the models, firing associations of HD cells during exploratory behavior are maintained during sleep. In addition, reliable spike transmission is observed between HD cells in the anterodorsal thalamic nucleus and the presubiculum (*Peyrache et al., 2015*), suggesting that cortical cells inherit their preferred HD from thalamic HD cells.

All models of HD cells require a mechanism to anchor the preferred direction of HD cells to the external world, thereby preventing error accumulation inherent to head-movement integration (*Skaggs et al., 1995*; *Redish et al., 1996*; *Zhang, 1996*; *Bicanski and Burgess, 2016*). In these models, the anchoring mechanism involves the interaction between HD cells and cells encoding information about visual landmarks. Recent experimental work has shown that a subset of directional cells in the retrosplenial cortex are indeed controlled primarily by local visual landmarks, and that these cells can change their preferred directions independently of classic HD cells (*Jacob et al., 2017*). These observations suggest that cortical regions containing both classic HD cells and cells controlled by visual landmarks might be key sites where visual landmarks gain control over the head direction signal (*Goodridge and Taube, 1997*; *Clark et al., 2010*).

The medial entorhinal cortex and parasubiculum (MEC/PaS) are located at the top of hierarchical ascending HD pathways (*Clark and Taube, 2012*). There, HD cells intermingle with speed cells and grid cells (*Sargolini et al., 2006*; *Boccara et al., 2010*; *Kropff et al., 2015*). It has been proposed that a key function of these cells is to compute an estimate of the animal's position in space, which is optimally represented by the activity of modularly organized grid cells (*McNaughton et al., 2006*; *Stensola et al., 2012*; *Herz et al., 2017*). To keep track of the animal's position during movement, the grid cell network is thought to integrate direction and speed of movement using the activity of HD and speed cells, respectively (*McNaughton et al., 2006*; *Sargolini et al., 2006*; *Kropff et al., 2015*). Thus, this spatial code likely depends on the properties of its HD inputs (*Winter et al., 2015*).

The MEC/PaS also receive important visual and visuospatial inputs from the postrhinal and retrosplenial cortices (*Wyss and Van Groen, 1992*; *Burwell and Amaral, 1998*; *Czajkowski et al., 2013*; *Koganezawa et al., 2015*). Indeed, recent studies have established that a substantial fraction of MEC/PaS neurons encode information about visual patterns or contexts (*Pérez-Escobar et al., 2016*; *Diehl et al., 2017*; *Ismakov et al., 2017*). The presence of cells influenced by visual landmarks in the MEC/PaS raises the possibility that the MEC/PaS is a site of interaction between information about HD and visual cues. We therefore investigated whether some MEC/PaS neurons express a conjunctive code for HD and visual landmarks. We recorded the activity of HD cells in the MEC/PaS and tested whether their activity was also controlled by visual landmarks. The recording environment consisted of an elevated platform surrounded by four walls. Two distinct light patterns located on different walls were turned on and off. We report that HD cells in the MEC/PaS could be divided into two classes: non-rhythmic and theta-rhythmic HD cells. Changing which visual pattern was presented caused alterations in the tuning curves of both non-rhythmic and theta-rhythmic HD cells, but the magnitude of the changes was larger for non-rhythmic HD cells, with some non-rhythmic HD cells switching from active to nearly silent when the light pattern changed. Simultaneous recordings from multiple HD cells revealed that while theta-rhythmic HD cells maintained their firing associations when landmarks changed, the tuning changes of non-rhythmic HD cells were often non-coherent, demonstrating that non-rhythmic HD cells were not constrained by attractor-like dynamics but rather controlled by visual landmarks. We suggest that landmark-controlled non-rhythmic HD cells might contribute to setting the preferred directions of classic HD cells when an animal enters visually distinct environments.

# Results

To investigate the impact of visual landmarks on the HD signal, we recorded the activity of MEC/PaS neurons together with oscillatory field potentials using multichannel extracellular techniques in freely behaving mice. Nine mice were trained to explore an elevated square platform surrounded by four walls (*Figure 1a*). Two distinct visual patterns (vp1 and vp2) made from different arrangements of LED strips were attached to two adjacent walls. In addition, a standard cue card attached to one wall remained at the same location throughout the experiment (*Figure 1a*). The two distinct visual patterns were switched on and off in turn, without any interleaving dark periods. Recording sessions included forty 2-min trials alternating between vp1 and vp2 trials (*Figure 1b*).

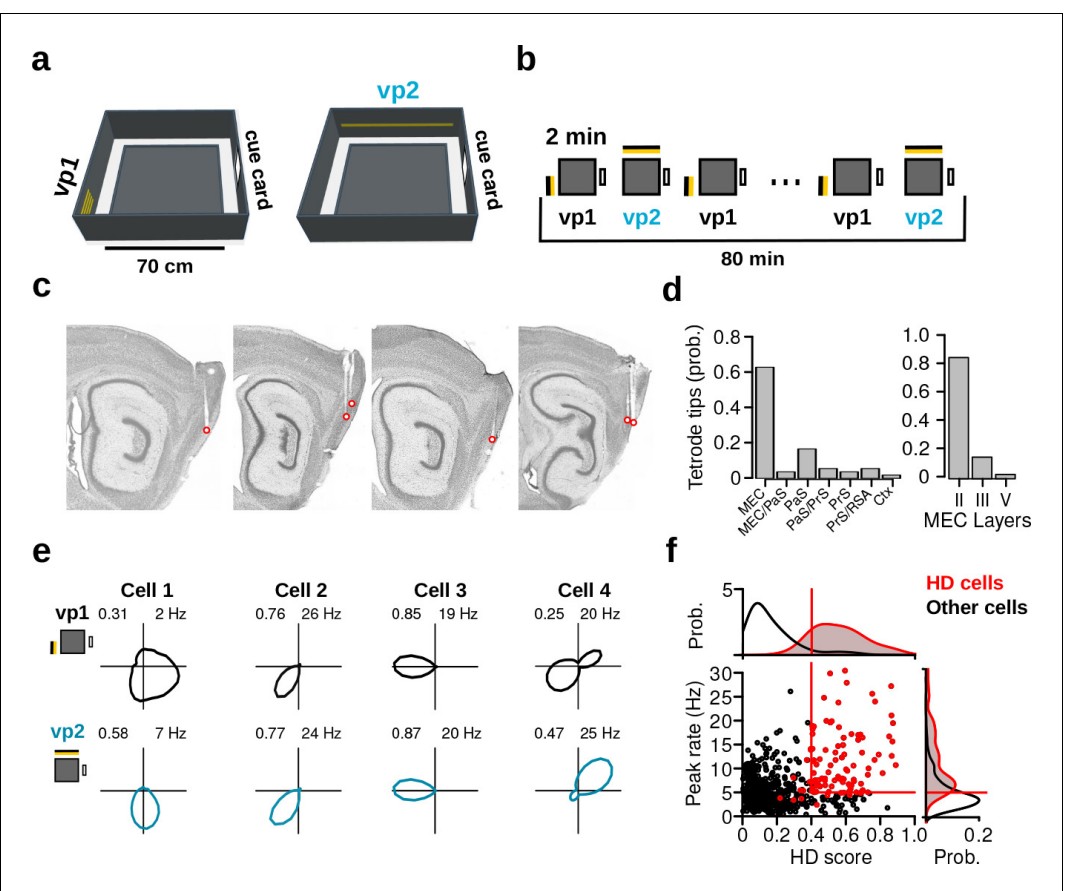

**Figure 1.** Recording protocol, histological results and examples of HD cells. (**a**) The recording environment was an elevated square platform surrounded by four walls. Two distinct visual patterns (vp1 and vp2) made of LED strips were attached to two adjacent walls. A standard paper cue card was attached to a third wall. (**b**) Recording sessions comprised a sequence of forty 2-min trials that alternated between vp1 and vp2 trials. (**c**) Sagittal brain sections showing representative recording sites in the MEC and PaS. Red circles indicate tetrode tips. (**d**) Distribution of tetrode tips across brain regions and different layers. MEC: medial entorhinal cortex, PaS: parasubiculum, PrS: Presubiculum, RSA: retrosplenial agranular cortex, Ctx: cortex. (**e**) HD firing rate polar plots for four HD cells recorded during the two light conditions (numbers indicate HD score and peak firing rate). (**f**) Scatter plot showing HD scores and peak firing rates of all neurons during vp2 trials. Each dot represents one cell. Lines indicate thresholds for HD cells identification. Red dots are HD cells.

DOI: https://doi.org/10.7554/eLife.35949.002

The following figure supplements are available for figure 1:

**Figure supplement 1.** Examples of recording sites.
DOI: https://doi.org/10.7554/eLife.35949.003

**Figure supplement 2.** Directional distributive ratio, HD cell properties and behavior during vp1 and vp2 trials.
DOI: https://doi.org/10.7554/eLife.35949.004

Histological analysis revealed that the final locations of most recording sites were in the MEC (63.0%, 34 out of 54; *Figure 1c–d*, *Figure 1—figure supplement 1*). Half of the remaining recording sites were found in the PaS (16.7%, 9 out of 54). Of the recording sites in the MEC, 82.4% (28 out of 34) had entered layer II of the MEC before the end of the experiment (*Figure 1d*). The final location of all visible tetrode tips is presented in *Supplementary file 1*.

A total of 944 neurons were recorded over 167 recording sessions. The number of cells recorded in each animal is presented in *Supplementary file 1*. The HD tuning curve of each neuron was calculated separately for trials with vp1 and vp2 (*Figure 1e*). The HD score, which was defined as the mean vector length of the tuning curve, served as a measure of HD selectivity. Cells with a HD score exceeding 0.4 and a peak firing rate larger than 5 Hz during vp1 or vp2 trials were considered putative HD cells (106 out of 944 neurons, *Figure 1f*). To ensure that HD selectivity was not a byproduct of spatial selectivity coupled with unequal HD sampling across the recording environment, we calculated a directional distributive ratio for each HD cell (*Muller et al., 1994*; *Cacucci et al., 2004*) (Materials and methods and *Figure 1—figure supplement 2a–c*). A directional distributive ratio approaching 0 indicated that the observed HD tuning curve could result from spatial selectivity coupled with biased HD sampling. Only HD cells with a directional distributive ratio larger than 0.2 were included in further analysis (*N* = 104). Out of 104 HD cells, 10.6% (*N* = 11) were conjunctive grid x HD cells and were excluded from the HD cell category, leaving a total of 93 HD cells. Of the remaining HD cells, 29.0% were speed modulated, 62.3% had significant spatial sparsity scores, and 31.2% were HD selective only (*Figure 1—figure supplement 2d*).

HD cells expressed similar levels of directional tuning during vp1 and vp2 trials. At the population level, HD scores, peak firing rates and mean firing rates of HD cells were not significantly different between the two trial types (*Figure 1—figure supplement 2f*). Preferred directions of HD cells were randomly distributed during both trial types (Rayleigh test of uniformity, vp1: test statistic = 0.0931, p=0.4466; vp2: test statistic = 0.0479, p=0.81). Moreover, the behavior of mice was comparable between the two different trial types (*Figure 1—figure supplement 2g*).

## Theta rhythmicity reveals two HD cell populations

A substantial proportion of MEC/PaS neurons shows prominent rhythmic activity in the theta frequency band (6–10 Hz) (*Alonso and García-Austt, 1987*; *Cacucci et al., 2004*; *Mizuseki et al., 2009*). Surprisingly, we observed a large variability in the theta rhythmic activity of HD cells, with a group of HD cells firing rhythmically at theta frequency (*Figure 2a*) and a second group showing no clear rhythmic activity in this frequency range (*Figure 2b*).

To quantify this observation, we first calculated for each neuron a power spectrum from the instantaneous firing rate. We found a large variability in the power at theta frequency between neurons. A theta index was calculated by comparing the power in the theta frequency band (6–10 Hz) to that of two adjacent frequency bands (3–5 and 11–13 Hz). The distribution of theta indices for all HD cells is shown in *Figure 3a*. The theta index distribution was best fit by a model with two components (see Materials and methods; Gaussian finite mixture models: log-likelihood = 98.25, *df* = 5, BIC = 173.83). In line with Alonso and García-Austt's work (*Alonso and García-Austt, 1987*), these two components corresponded to non-rhythmic and theta-rhythmic cells (*Figure 3a* and *Figure 3—figure supplement 1a*, *N* = 34 non-rhythmic, *N* = 59 theta-rhythmic, theta index threshold = 0.07). Similar results were obtained when the lower adjacent frequency band used to calculate the theta index was lowered to 1.5–3 Hz in order to minimize the influence of subharmonic theta.

We investigated the relationship between HD cell firing activity and theta oscillations of the local field potentials. For each cell, the firing probability as a function of theta phase, together with the mean vector length and the preferred theta phase of its spikes were calculated. Spikes of theta-rhythmic HD cells had larger mean vector lengths than non-rhythmic HD cells (*Figure 3b*, left; Wilcoxon rank-sum test, $w = 19$, $P < 10^{-15}$). Theta-rhythmic HD cells fired preferentially at the end of the descending phase of theta oscillations (*Figure 3b*, right; mean preferred phase: 59.27°; Rayleigh test of uniformity, test statistic = 0.2685, p $< 10^{-19}$), whereas non-rhythmic HD cells did not show such preference (test statistic = 0.2685, p=0.09). Theta phase modulated the firing rate of all theta-rhythmic HD cells (Rayleigh test of uniformity). Even though non-rhythmic HD cells displayed weak theta phase modulation, the firing rate of 94.1% (32 out of 34) of non-rhythmic HD cells was still significantly modulated by theta phase.

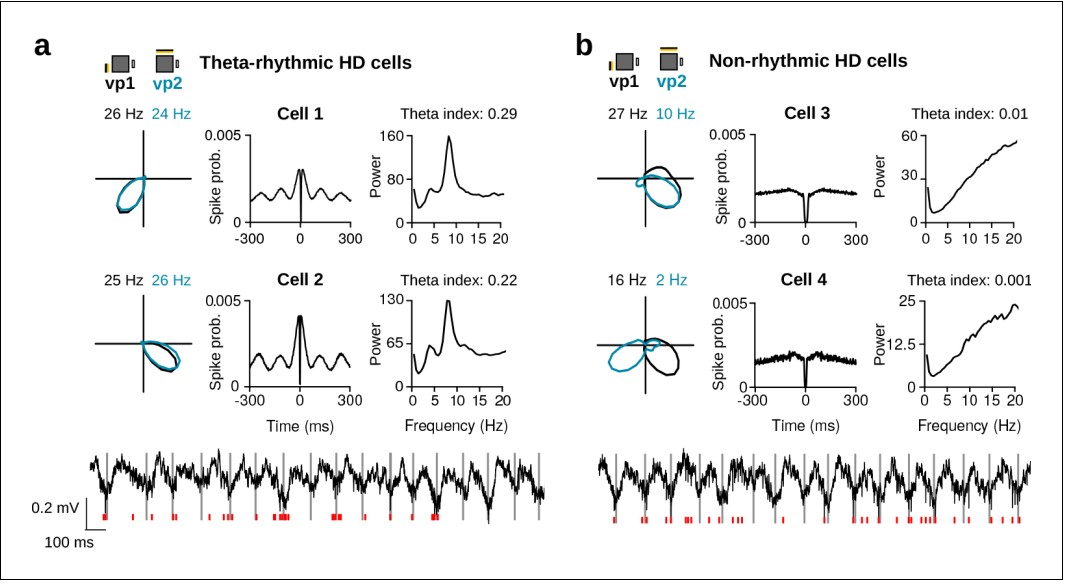

**Figure 2.** Variability in theta rhythmic activity of HD cells. (a) Examples of two theta-rhythmic HD cells (Cell 1 and Cell 2) and (b) two non-rhythmic HD cells (Cell 3 and Cell 4). From left to right for each cell: HD tuning curves during vp1 and vp2 trials (numbers indicate peak firing rates), spike-time autocorrelation, and power spectrum of the instantaneous firing rate. Bottom: raw signals with spike times (red vertical tics) of Cell 2 and Cell 4 shown above. Vertical gray lines are aligned to the troughs of theta cycles.
DOI: https://doi.org/10.7554/eLife.35949.005

These substantial differences in theta modulation were not caused by differences in the local field potential theta power recorded in the proximity of the cell bodies. For each HD cell, we calculated the power spectrum from the local field potentials of its respective tetrode. The power at theta frequency in the local field potentials was similar for non-rhythmic and theta-rhythmic HD cells (*Figure 3—figure supplement 1b*, Wilcoxon rank-sum test, peak theta power: $w = 1024.5$, p=0.87). There was no significant difference between theta indices of cells recorded in hemispheres in which all tetrode tips were located in the MEC compared to those in which some tetrode tips were in the PaS ($w = 2433.5$, p=0.75).

The HD selectivity and mean firing rate of non-rhythmic and theta-rhythmic HD cells during vp1 trials were similar (*Figure 3c*, Wilcoxon rank-sum test, HD score: $w = 881$, p=0.33, mean firing rate: $w = 993$, p=0.94), demonstrating that theta-rhythmic and non-rhythmic HD cells displayed similar levels of directional tuning.

Theta-cycle skipping was observed in a minority of theta-rhythmic HD cells. When using a theta-cycle skipping index of 0.1 as threshold (see Materials and methods), 10% (6 out of 59) of theta-rhythmic HD cells showed theta-cycle skipping. This was slightly lower than what had been previously reported in rats (*Brandon et al., 2013*).

We then investigated whether non-rhythmic and theta-rhythmic HD cells form anatomically segregated populations within the MEC/PaS region. By analyzing pairs of HD cells simultaneously recorded on the same tetrode ($N = 17$), we found only one mixed non-rhythmic/theta-rhythmic pair, suggesting that the two cell populations were partially anatomically segregated. Also, fewer non-rhythmic HD cells were recorded on the same tetrode as grid cells compared to theta-rhythmic HD cells (*Figure 3—figure supplement 1c*, $N = 9$ and 42, respectively) (Chi-squared test: $\chi^2 = 15.66$, $df = 1$, p $<10^{-5}$). Finally, non-rhythmic HD cells tended to be recorded during earlier recording sessions compared to theta-rhythmic HD cells (*Figure 3d*, $w = 497$, p $<10^{-5}$), suggesting that non-rhythmic HD cells were located closer to the dorsal border of the MEC/PaS or in deeper layers.

To confirm the anatomical localization of non-rhythmic HD cells, we implanted four additional mice in which the minimal spacing between tetrodes was increased to approximately 0.5 mm, making it possible to assign each cell to a tetrode track. These mice were trained on a shorter recording protocol (60 min per session instead of 80 min) and perfused with the tetrode left in place as soon

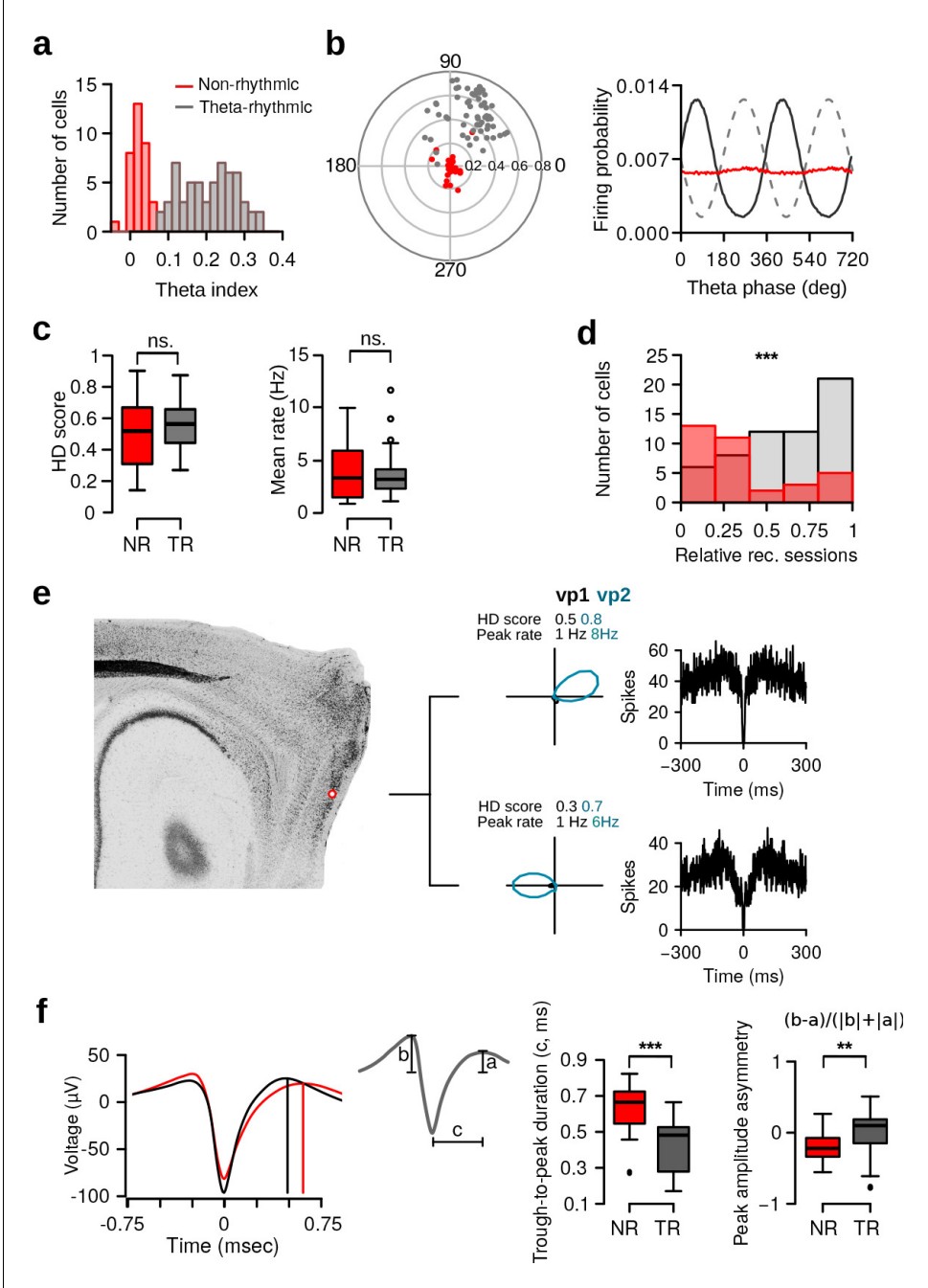

**Figure 3.** Theta rhythmic activity identifies two populations of HD cells. (a) Distribution of theta indices of all HD cells. The two populations were identified with a Gaussian finite mixture model. Red: non-rhythmic cells, gray: theta-rhythmic cells. (b) Preferred theta phase and phase locking of HD cells to the LFP theta oscillations. Left: Polar plot showing the preferred theta phase and theta phase modulation of HD cells. Non-rhythmic and theta-rhythmic HD cells are depicted in red and gray, respectively. Right: Mean firing probability as a function of theta phase of non-rhythmic (red) and theta-rhythmic (black) HD cells, the 90° phase represents the trough of the theta cycle (dashed line). (c) HD scores and mean firing rates of non-rhythmic (NR, red) and theta-rhythmic (TR, gray) HD cells during vp1 trials. (d) Relative recording sessions during which non-rhythmic and theta-rhythmic HD cells were recorded. A score of 0 and 1 indicate that the cell was recorded on the first and last recording session of an animal, respectively. (e) Tetrode tracks from a mouse perfused immediately after recording two non-rhythmic HD cells. The tetrode tips were located in the most dorsal portion of the MEC. The HD tuning curve during vp1 and vp2 trials and the spike-time autocorrelation are shown for each cell. (f) Mean spike waveform (left), trough-to-peak duration (middle), and peak amplitude asymmetry (right) of non-rhythmic and theta-rhythmic HD cells. Cells which

*Figure 3 continued on next page*

*Figure 3 continued*

had inverted spike waveforms (N = 4 non-rhythmic and 7 theta-rhythmic) were excluded from this analysis. ns.: not significant, **: p < 0.01, ***: p < 0.001.

DOI: https://doi.org/10.7554/eLife.35949.006

The following figure supplements are available for figure 3:

**Figure supplement 1.** Properties of theta-rhythmic and non-rhythmic HD cells.

DOI: https://doi.org/10.7554/eLife.35949.007

**Figure supplement 2.** Non-rhythmic HD cells of the parahippocampal formation.

DOI: https://doi.org/10.7554/eLife.35949.008

as non-rhythmic HD cells were recorded. Out of the seven recording sites with non-rhythmic HD cells that were recovered, four were in the MEC (*Figure 3e*). Interestingly, three recording sites were in the postrhinal cortex (*Figure 3—figure supplement 2*), suggesting that the postrhinal cortex might also contain non-rhythmic HD cells. Note that stopping this experiment as soon as non-rhythmic HD cells were encountered biased our sampling toward more dorsal locations.

Analysis of the spike waveforms in the two populations revealed that non-rhythmic HD cells had longer trough-to-peak durations (*Figure 3f*, N = 30 and 52, w = 1347, p $<10^{-8}$) and larger spike asymmetry, compared to theta-rhythmic HD cells (*Figure 3f*, w = 463, p=0.002). Taken together, these results demonstrate that HD cells in the parahippocampal region can be divided into non-rhythmic and theta-rhythmic HD cells. These two cell types appear partially anatomically segregated and have distinct intrinsic biophysical properties.

## Visual landmarks alter the tuning curves of HD cells

Having established that HD cells of the MEC/PaS form two populations, we asked whether HD cells responded to changes in visual landmarks. Inspection of the tuning curves during vp1 and vp2 trials suggested that the preferred direction of some HD cells changed with the different visual patterns (*Figure 4a*). For some HD cells, these changes could be observed between consecutive trials, causing the preferred direction of the neurons to oscillate in time (*Figure 4a*). To test whether the change in preferred direction of each cell was statistically significant, we compared the observed change to a surrogate distribution of changes obtained by randomly reassigning trial labels (vp1 and vp2) (*Figure 4a*, see Materials and methods). Observed changes that were larger than 99% of the surrogate changes were considered significant. We found that 63.4% (59 out of 93) of the HD cells significantly altered their preferred HD between vp1 and vp2 trials.

A similar analysis was performed to test whether the level of directionality of HD cells changed with the visual patterns presented to the animal (*Figure 4b*). We observed HD cells that changed their selectivity depending on the visual patterns (*Figure 4b*). These changes could often be observed between single trials (*Figure 4b*). Overall, 47.3% (44 out of 93) of the HD cells significantly altered their HD score between trial types.

We next compared the responses of non-rhythmic and theta-rhythmic HD cells to visual landmark manipulations. Non-rhythmic HD cells showed larger changes in preferred HD between vp1 and vp2 trials than theta-rhythmic HD cells (*Figure 4c*, w = 1465, p=0.00012). Likewise, non-rhythmic HD cells also showed larger changes in HD selectivity (*Figure 4d*, w = 1618, p $<10^{-7}$). This indicates that visually driven changes in tuning curves were more pronounced in non-rhythmic HD cells compared to theta-rhythmic HD cells.

It was previously shown that HD cells located in the presubiculum and anterodorsal thalamic nucleus maintained their firing rate when an animal explored different environments (*Goodridge et al., 1998*; *Taube and Burton, 1995*; *Taube et al., 1990b*). We therefore tested whether the firing rate of HD cells in the MEC/PaS was preserved when visual landmarks were manipulated. We observed HD cells with pronounced alterations in their firing rate with some cells being almost silent in response to one of the two different visual patterns (*Figure 5a*). For these cells, the instantaneous firing rate oscillated in time, in line with the trial type (*Figure 5a*). To test whether these changes were significant, we calculated the relative change in firing rate for each neuron: $\left|\frac{r_{vp1}-r_{vp2}}{r_{vp1}+r_{vp2}}\right|$. We found that 40.9% (38 out of 93) of the HD cells significantly altered their mean firing rate in response to changes in visual landmarks. Thus, these results demonstrate that the mean

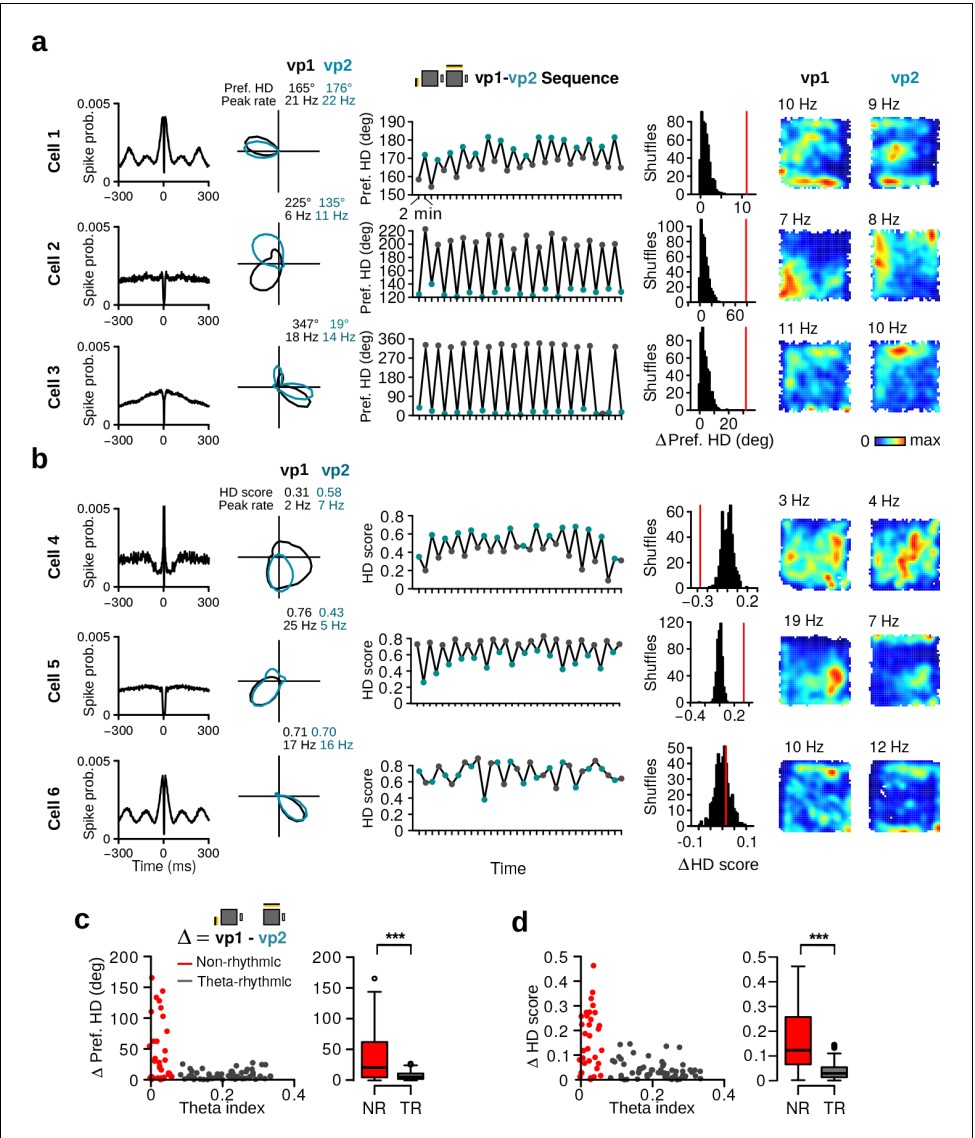

**Figure 4.** Alteration of preferred HD and HD selectivity following visual landmark manipulation. (a) Examples of three HD cells that changed their preferred direction between vp1 and vp2 trials. One cell per row. From left to right: Spike-time autocorrelation, HD tuning curves during vp1 (black) and vp2 (blue) trials, preferred HD during individual 2-min trials, observed change in preferred HD between vp1 and vp2 trials (red line) with distribution of preferred HD changes when trial labels were reassigned randomly, and firing rate maps during vp1 and vp2 trials. Numbers above the firing rate maps are peak firing rates. (b) Examples of three HD cells with different HD selectivity during vp1 and vp2 trials. From left to right: Same as in a) but the third and forth columns show HD scores instead of preferred directions. (c) Left: Scatter plot of the theta index of each HD cell against its change in preferred HD. Right: Change in preferred HD between vp1 and vp2 trials for non-rhythmic and theta-rhythmic cells. (d) Same as (c) but for HD scores. ***: p < 0.001.

DOI: https://doi.org/10.7554/eLife.35949.009

The following figure supplements are available for figure 4:

**Figure supplement 1.** Distribution of changes in HD tuning.
DOI: https://doi.org/10.7554/eLife.35949.010

**Figure supplement 2.** Properties of HD cells changing their preferred HD between vp1 and vp2 trials (N = 59 cells).
DOI: https://doi.org/10.7554/eLife.35949.011

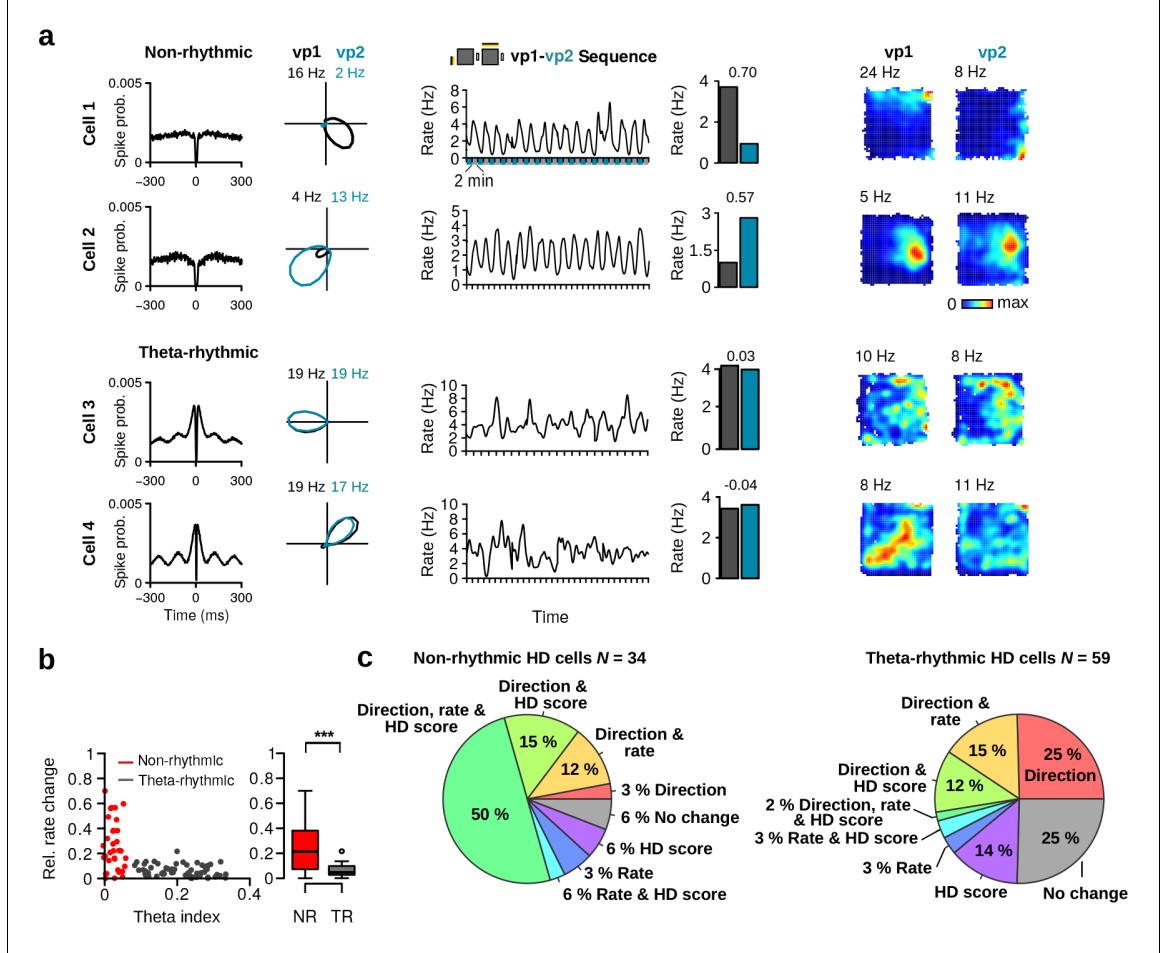

**Figure 5.** Firing rate changes induced by visual landmark manipulation. (a) For each cell from left to right: Spike-time autocorrelation, HD tuning curves during vp1 (black) and vp2 (blue) trials, instantaneous firing rates (standard deviation Gaussian kernel 25 s, window size 1 s) as a function of time, mean firing rate during vp1 and vp2 trials, and firing rate maps during vp1 and vp2 trials. Numbers above the histograms indicate the relative change in rate. Numbers above the firing rate maps are peak firing rates. (b) Left: Scatter plot of the theta index of each HD cell against its relative firing rate change across trial types. Right: Relative firing rate change between vp1 and vp2 trials for non-rhythmic and theta-rhythmic HD cells. (c) Pie charts illustrating the percentages of non-rhythmic (left) and theta-rhythmic (right) HD cells with significant changes in preferred direction, HD score or mean firing rate. ***: $P < 0.001$.

DOI: https://doi.org/10.7554/eLife.35949.012

The following figure supplement is available for figure 5:

**Figure supplement 1.** Non-rhythmic HD cells appear to encode allocentric HD rather than egocentric landmark bearing.

DOI: https://doi.org/10.7554/eLife.35949.013

firing rate of HD cells in the MEC/PaS is modulated by visual landmarks. The magnitude of the firing rate changes was significantly larger for non-rhythmic than for theta-rhythmic HD cells (*Figure 5b*, $w = 1641$, $p < 10^{-7}$).

*Figure 5c* summarizes the different types of changes observed in the non-rhythmic and theta-rhythmic HD cell populations. The percentage of cells with a significant visually driven change was 94.1% and 74.5% for non-rhythmic and theta-rhythmic cells, respectively (*Figure 5c*, $\chi^2 = 3.4465$, $df = 1$, $p=0.06$). Taken together, these results show that the tuning curves of MEC/PaS HD cells are affected by the visual landmarks. The tuning curves of non-rhythmic HD cells showed larger landmark-driven changes than those of theta-rhythmic HD cells.

The majority of neurons in the MEC are spatially selective (*Diehl et al., 2017*). It could therefore be argued that the changes in HD tuning between vp1 and vp2 trials were caused by altered spatial firing patterns of HD cells. To rule out this possibility, we focused our analysis on a subset of HD cells

with no significant spatial modulation (non-significant sparsity score; 36 out of 93 HD cells). Within this population of non-spatial HD cells, 86.1% (31 out of 36) had a significant change in preferred direction, HD score or mean firing rate between the vp1 and vp2 trials. This proportion was comparable to that observed when considering spatially selective HD cells (Pearson's Chi-squared test: $\chi^2$ = 0.3543, $df$ = 1, p=0.55). Thus, the changes in HD tuning curves did not result from an altered spatial signal between vp1 and vp2 trials.

We also tested whether a similar proportion of HD cells showed landmark-driven changes when limiting the analysis to cells recorded from hemispheres in which all tetrode tips were positioned in the MEC. We found that 83.6% (56 out of 67) of the MEC HD cells displayed significant changes in preferred direction, HD selectivity or mean firing rate between trial types. A similar percentage was obtained when analyzing HD cells from hemispheres containing recording sites in the PaS, presubiculum or cortex (76.9%, 20 out of 26 cells, $\chi^2$ = 0.1996, $df$ = 1, p=0.66).

It was previously shown that the width of a prominent visual landmark could affect the width of the HD tuning curves of hippocampal neurons during navigation in a virtual environment (*Acharya et al., 2016*). In the same study, a prominent visual landmark could also bias the distribution of preferred directions toward the landmark. To address whether these two effects could contribute to the differences observed between vp1 and vp2 trials, we analyzed selectively HD cells that showed a tuning curve change between vp1 and vp2 trials. No difference in HD scores was observed between vp1 and vp2 trials (*Figure 4—figure supplement 2a*). Moreover, there was no clear indication of an accumulation of preferred directions toward either vp1 or vp2 (*Figure 4—figure supplement 2b*).

Finally, one explanation of the strong response of non-rhythmic HD cells to landmark manipulations is that these cells might encode egocentric landmark bearing information instead of allocentric HD. Egocentric landmark bearing cells fire when a landmark is located at a given position in the animal's field of view (*Bicanski and Burgess, 2016*). If a HD cell is an egocentric landmark bearing cell, its preferred HD should change systematically as a function of the animal's position within the arena due to the parallax effect. We could not detect clear evidence for such systematic changes in HD as a function of position in non-rhythmic HD cells (see *Figure 5—figure supplement 1*).

## Bidirectionality of HD cells is affected by visual landmarks

Inspection of the HD cell tuning curves also revealed that a subpopulation of HD cells had two preferred directions (*Figure 6a*, see also *Figure 1e*-Cell-4, *Figure 4a*-Cell-3). Some of these tuning curves were similar to those previously reported for the retrosplenial cortex (*Jacob et al., 2017*), but the two peaks were not always at an 180° angle to each other. The bidirectionality of these cells often varied with the trial types (*Figure 6a*).

To quantify these observations, a bidirectionality (BD) score was defined as the firing rate ratio of the two largest peaks in the tuning curve (see Materials and methods). For some cells, the BD score changed on a trial-by-trial basis (*Figure 6a*). The bidirectionality in the tuning curves could be observed over short periods of 30 s (*Figure 6b*), suggesting that bidirectionality was not caused by inter-trial instability of a single preferred direction.

Bidirectional HD cells were defined as HD cells for which the BD score was larger than 0.2, the peak firing rates of the two peaks were larger than 2 Hz, and the ratio between the firing rate of the second largest peak and the trough in the HD tuning curve with the highest firing rate between the peaks exceeded 1.25. These criteria had to be reached during vp1 or vp2 trials. Note that a bidirectional cell with equally dominant peaks at 180 degrees to each other during both trial types would not have been classified as HD cell in either trial type and, therefore, would not have been included in the analysis.

The analysis showed that 15.1% (14 out of 93, eight non-rhythmic and six theta-rhythmic) of the HD cells were classified as bidirectional, with two cells being bidirectional only during vp1 trials, and 6 cells only during vp2 trials. The remaining six cells were bidirectional during both trial types. The distribution of angles between the two preferred directions of bidirectional HD cells had two modes, close to 90° and 180 degrees (*Figure 6c*). In some HD cells, the changes in HD score and preferred direction between vp1 and vp2 trials could be explained by a change in bidirectionality (*Figure 6a–b*). The changes in the BD scores between vp1 and vp2 trials were larger than chance levels (*Figure 6d*; paired Wilcoxon signed-rank test, N = 14, v = 12, p=0.009). Overall, non-rhythmic HD

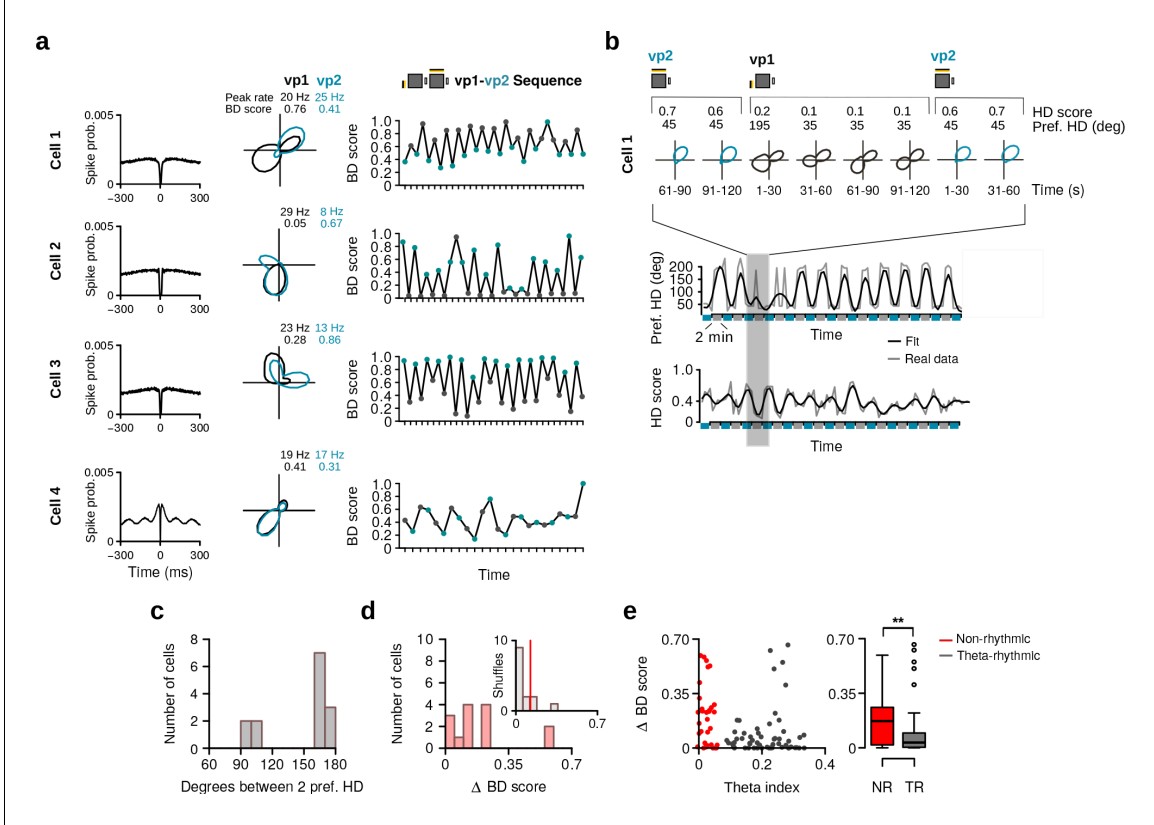

**Figure 6.** HD cells with bidirectional tuning curves. (**a**) Four HD cells with bidirectional HD tuning curves showing changes with the two trial types. From left to right: spike-time autocorrelations, tuning curves during vp1 and vp2 trials, with numbers indicating the peak firing rates and bidirectionality (BD) scores during vp1 and vp2 trials, and evolution of the BD score during the recording session. (**b**) Example of a bidirectional HD cell during the transition from vp2 to vp1 trials at a shorter time scale. Top panel: polar plots during 30 s blocks showing that the two preferred directions were expressed within a single vp1 trial. Bottom panel: preferred HD and HD score of the cell over time. (**c**) Distribution of angles between the two preferred HDs of bidirectional HD cells. (**d**) Distribution of change in BD score between vp1 and vp2 trials for HD cells with a bidirectional HD tuning curve (red, $N = 14$). Inset: the median of observed changes in BD score (red line) with the distribution expected by chance (gray). (**e**) Left: Scatter plot of the theta index against the change in BD score of each HD cell. Right: Change in BD score between vp1 and vp2 trials for non-rhythmic and theta-rhythmic cells. **: $p < 0.01$.

DOI: https://doi.org/10.7554/eLife.35949.014

The following figure supplement is available for figure 6:

**Figure supplement 1.** Cells with bidirectional HD tuning curves that were not classified as HD cells ($N = 36$).

DOI: https://doi.org/10.7554/eLife.35949.015

cells showed larger changes in their BD score compared to theta-rhythmic cells (*Figure 6e*, $w = 1366$, p=0.004). Including bidirectional cells that were not classified as HD cells ($N = 36$) did not affect this conclusion (*Figure 6—figure supplement 1*).

## Non-rhythmic HD cells reorganize their firing associations

Computational models of HD cells predict that differences in the preferred directions of HD cells are preserved at all times. Support for these models comes from numerous studies showing coherent rotation of the preferred directions of HD cells in response to landmark manipulations (*Taube et al., 1990b*; *Yoganarasimha et al., 2006*). However, these findings were obtained from recordings in subcortical areas or in the presubiculum, and it is still not known whether HD cells in the MEC/PaS display the coherence predicted by computational models. We therefore tested whether visually driven tuning curve changes observed in our protocol were coherent across simultaneously recorded HD cells.

We first inspected the tuning curves of simultaneously recorded HD cells ($N = 37$ HD cell pairs). *Figure 7a* shows examples of HD cells for which changes from vp1 to vp2 were not coherent. For

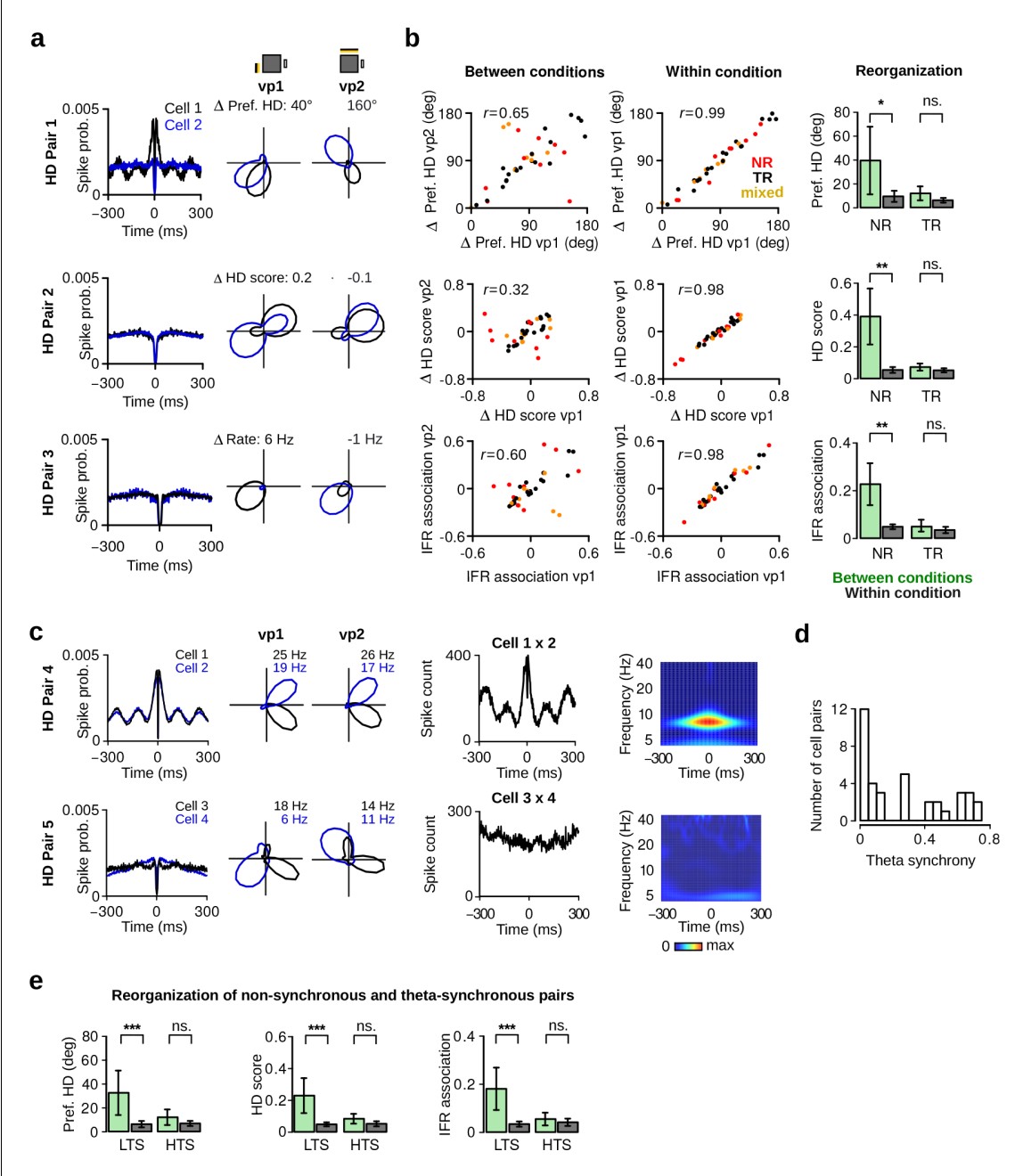

**Figure 7.** Reorganization of HD cells caused by changes in visual landmarks. (**a**) Three pairs of HD cells recorded simultaneously showing non-coherent changes. From left to right: Spike-time autocorrelations, HD polar plots of HD cell pairs during vp1 and vp2 trials. Numbers above the polar plots indicate the difference in preferred HD (HD Pair 1), in HD score (HD Pair 2) or in mean firing rate (HD Pair 3). (**b**) Left: correlation between the differences in preferred HD, in HD score and in instantaneous firing rate (IFR) association of HD cell pairs for trials with the same or different visual patterns. Data are shown for vp1 and vp2 trials (left, between conditions) or for two mutually exclusive subsets of vp1 trials (right, within condition). *r* values are correlation coefficients. Red, black and yellow data points represent non-rhythmic, theta-rhythmic and mixed cell pairs, respectively. Right: reorganization of preferred HD, HD score and IFR association of non-rhythmic (NR) and theta-rhythmic (TR) HD cell pairs between conditions or within condition. Plots show mean ±95% confidence intervals. (**c**) From left to right: spike-time autocorrelations, polar plots, spike-time crosscorrelations and wavelet transforms of the z-score-based spike-time crosscorrelations. Both wavelet transforms were normalized to the same scale from minimal power (dark blue) to maximal power (dark red). Wavelet transforms reveal high theta frequency synchronization for one of the HD cell pairs (HD Pair 4). (**d**) Distribution of theta synchrony scores for all HD cell pairs. (**e**) Reorganization for HD cell pairs with low (LTS) or high (HTS) theta synchrony. ns.: not significant, *: p < 0.05, **: p < 0.01, ***: p < 0.001.
DOI: https://doi.org/10.7554/eLife.35949.016

example, two HD cells modified their difference in preferred direction depending on which visual pattern was presented (*Figure 7a*, HD Pair 1). A similar uncoupling of simultaneously recorded HD cell pairs was also observed for HD selectivity (*Figure 7a*, HD Pair 2) and mean firing rate (*Figure 7a*, HD Pair 3).

To quantify these non-coherent changes in HD cell activity, we correlated the differences in preferred direction of HD cell pairs observed during vp1 and vp2 trials (between conditions) (*Figure 7b*, top left panel). As a control, we calculated the correlation coefficients of the same scores obtained from two mutually exclusive subsets of vp1 trials (within condition) (*Figure 7b*, top right panel). The correlation coefficient between conditions ($r = 0.65$) was significantly lower than that observed within condition ($r = 0.99$; difference, Fisher Z-transformation: Z-score = 7.72, p $<10^{-14}$). Similarly, differences in HD scores were less correlated between conditions than within condition (*Figure 7b*, middle panels; between conditions: $r = 0.32$, within condition: $r = 0.98$, difference: Z-score = 8.11, p $<10^{-16}$). We also assessed the firing associations of HD cells in the time domain. The instantaneous firing rate (IFR) of each HD cell was calculated and IFR association was defined as the correlation coefficient between IFR vectors of two HD cells. The stability of IFR associations was lower between conditions ($r = 0.60$) than within condition ($r = 0.98$) (*Figure 7b*, bottom panels; difference: Z-score = 6.62, p=10 $<10^{-11}$).

To further quantify non-coherent changes within the HD cell population, a reorganization score for preferred HD, HD score and IFR association was computed. After obtaining differences in preferred HD, HD score or IFR association for each cell pair, the absolute difference between the values from vp1 and vp2 trials or from two mutually exclusive subsets of vp1 trials served as reorganization score between or within condition, respectively (see Materials and methods). We compared reorganization scores of HD cell pairs formed by two theta-rhythmic or two non-rhythmic HD cells (*Figure 7b*). We found that theta-rhythmic HD cell pairs did not show significant reorganization when visual landmarks were altered (*Figure 7b*, N = 20, preferred HD: $v = 65$, p=0.14, HD score: $v = 72$, p=0.23, IFR association: $v = 69$, p=0.19). In contrast, visual landmark manipulations caused a reorganization of firing associations in non-rhythmic HD cell pairs (N = 9, preferred HD: $v = 2$, p=0.01, HD score: $v = 1$, p=0.00781, IFR association: $v = 0$, p=0.00391). These results indicate that the reorganization of firing associations was specific to non-rhythmic HD cells. Thus, the activity of HD cells is not entirely constrained by attractor-like dynamics.

We also tested whether visually driven reorganization between HD cells can be predicted based on their synchronous activity at theta frequency. We obtained time-frequency power spectra from spike-time crosscorrelations of simultaneously recorded HD cell pairs (*Peyrache et al., 2015*) (*Figure 7c*). Theta synchrony was defined as the average power in the time widow of 0±25 ms at theta frequency (6–10 Hz). From this, the distribution of theta synchrony scores was used to identify low-theta synchronous (LTS) and high-theta synchronous (HTS) HD cell pairs (*Figure 7d*, LTS < 0.2, N = 19, HTS > 0.2, N = 18). HTS HD cell pairs did not show significant visually-driven reorganization (*Figure 7e*, preferred HD: $v = 69$, p=0.50, HD score: $v = 51$, p=0.14, IFR association: $v = 67$, p=0.44). In contrast, LTS cell pairs showed significant reorganization when visual landmarks were altered (preferred HD: $v = 14$, p=0.00042, HD score: $v = 16$, p=0.00065, IFR association: $v = 18$, p=0.00097). These results show that HD cells which synchronized at theta frequency did not undergo visually driven reorganization. In contrast, non-synchronous HD cells reorganized their firing associations based on visual landmarks.

## Stable firing associations between grid cells

The above results indicate that one population of HD cells in the MEC/PaS shows visually-driven reorganization of their firing associations. This contradicts the prediction of current attractor network models of HD cells which states that the firing associations between HD cells are static. Another cell population in this brain region that is thought to be governed by attractor-like dynamics is that of grid cells (*McNaughton et al., 2006*). For this reason, we sought to determine whether the activity of simultaneously recorded grid cells remained coherent when visual landmarks were manipulated.

We recorded the activity of 219 grid cells during vp1 and vp2 trials (*Figure 8a* and *Figure 8—figure supplement 1a*). To investigate whether grid cells altered their firing properties in response to manipulation of visual landmarks, we assessed the stability of their firing rate maps. The firing rate map similarity between the two trial types was significantly lower than chance level (*Figure 8b*;

paired Wilcoxon signed-rank test, $N = 219$, $v = 439$, p $<10^{-16}$). We found that 48.4% (106 out of 219) of the grid cells showed significant changes in map similarity, preferred HD, or average firing rate between the vp1 and vp2 trials (*Figure 8c*).

Next, we examined whether the grid cell changes were coherent across simultaneously recorded grid cells ($N = 223$ grid cell pairs, *Figure 8d*), as predicted by attractor network models. Firing associations based on instantaneous firing rates were highly preserved between vp1 and vp2 trials (*Figure 8e*, top left panel, $r = 0.96$). Likewise, pairwise map similarity from the vp1 and vp2 trials was also strongly correlated (*Figure 8e*, bottom left panel, $r = 0.95$). In addition, reorganization levels for pairs of grid cells and pairs of HD cells were compared (*Figure 8f*). Only HD cells showed significant reorganization of their IFR association and of their firing rate maps between different trial types (reorganization of grid cell pairs between vs. within condition for IFR associations: $v = 10911$, p=0.10; map similarity: $v = 10661$, p=0.058). Similar results were obtained when considering only grid cell pairs, with at least one grid cell showing significant visually-driven changes of either map similarity, average firing rate or preferred HD (*Figure 8—figure supplement 1b–c*, $N = 106$ grid cell pairs, IFR association: $v = 2296$, p=0.089; map similarity: $v = 2247$, p=0.064). These findings suggest that the firing associations of grid cells were not significantly altered in response to changes in visual landmarks.

## Discussion

HD cells of the MEC/PaS are at the top level of the HD circuit hierarchy. Because a substantial fraction of MEC/PaS neurons encode information about visual patterns or contexts (*Pérez-Escobar et al., 2016*; *Diehl et al., 2017*; *Ismakov et al., 2017*), we set out to test whether HD cells in the MEC/PaS express a conjunctive code for visual landmarks and HD. We found that HD cells could be divided into two functionally distinct classes based on whether they fired rhythmically at theta frequency. Theta-rhythmic and non-rhythmic HD cells showed significant changes in their tuning curves when visual landmarks were manipulated, but the magnitude of the changes was larger for non-rhythmic HD cells. Moreover, non-rhythmic HD cells displayed non-coherent responses to visual landmark manipulations, whereas theta-rhythmic HD cells showed stable firing associations.

These properties of MEC/PaS HD cells set them apart from classic HD cells. Indeed, in the antero-dorsal thalamic nucleus and presubiculum the peak firing rate and directional tuning of HD cells were unchanged when an animal explored visually distinct environments (*Taube et al., 1990b*; *Yoder et al., 2011*). These results suggest that, in these two brain regions, HD cells are driven mainly by self-motion cues derived from vestibular, proprioceptive, and motor efference information (*Stackman and Taube, 1997*; *Yoder and Taube, 2014*) and the principal effect of visual landmarks on HD cells is to anchor the HD signal to the external world (*Skaggs et al., 1995*; *Yoder and Taube, 2009*; *Yoder et al., 2011*). Our findings demonstrate that the impact of visual information on MEC/PaS HD cells extends beyond this anchoring mechanism. The firing rate of several non-rhythmic HD cells was strongly modulated by the two visual landmarks, with some cells being almost silent with one of the two landmarks. This indicates that some HD cells are driven by visual landmarks, rather than being simply anchored by them. We suggest that a significant proportion of HD cells in the MEC/PaS express a conjunctive code for HD, visual landmarks and position. The origin of this 2D visual landmark modulation of HD cell activity is still unknown. Two possible origins are other MEC neurons changing their firing patterns depending on visual landmarks (*Pérez-Escobar et al., 2016*; *Diehl et al., 2017*), or afferent connections from visual cortical areas (e.g., postrhinal or retrosplenial cortex) (*Burwell and Amaral, 1998*; *Czajkowski et al., 2013*; *Koganezawa et al., 2015*).

A core tenet of HD cell models is that cells within the network have a fixed connectivity and that differences in preferred direction between HD cells do not change (*Skaggs et al., 1995*; *Zhang, 1996*; *Redish et al., 1996*). Although the HD signal is generated at the level of the dorsal tegmental nucleus and the lateral mammillary nucleus, the evidence supporting this assertion has been obtained mainly from recordings performed in the anterodorsal thalamic nucleus, presubiculum or cortical areas (*Taube et al., 1990a*; *Stackman et al., 2003*; *Yoganarasimha et al., 2006*; *Peyrache et al., 2015*). This suggests that the attractor-like network dynamics of the HD signal generator propagate from subcortical networks to cortical areas via ascending excitatory connections (*Peyrache et al., 2015*). In the current study, we found that the firing pattern of non-rhythmic HD cells within the MEC/PaS is not strictly controlled by these attractor-like network dynamics. Indeed,

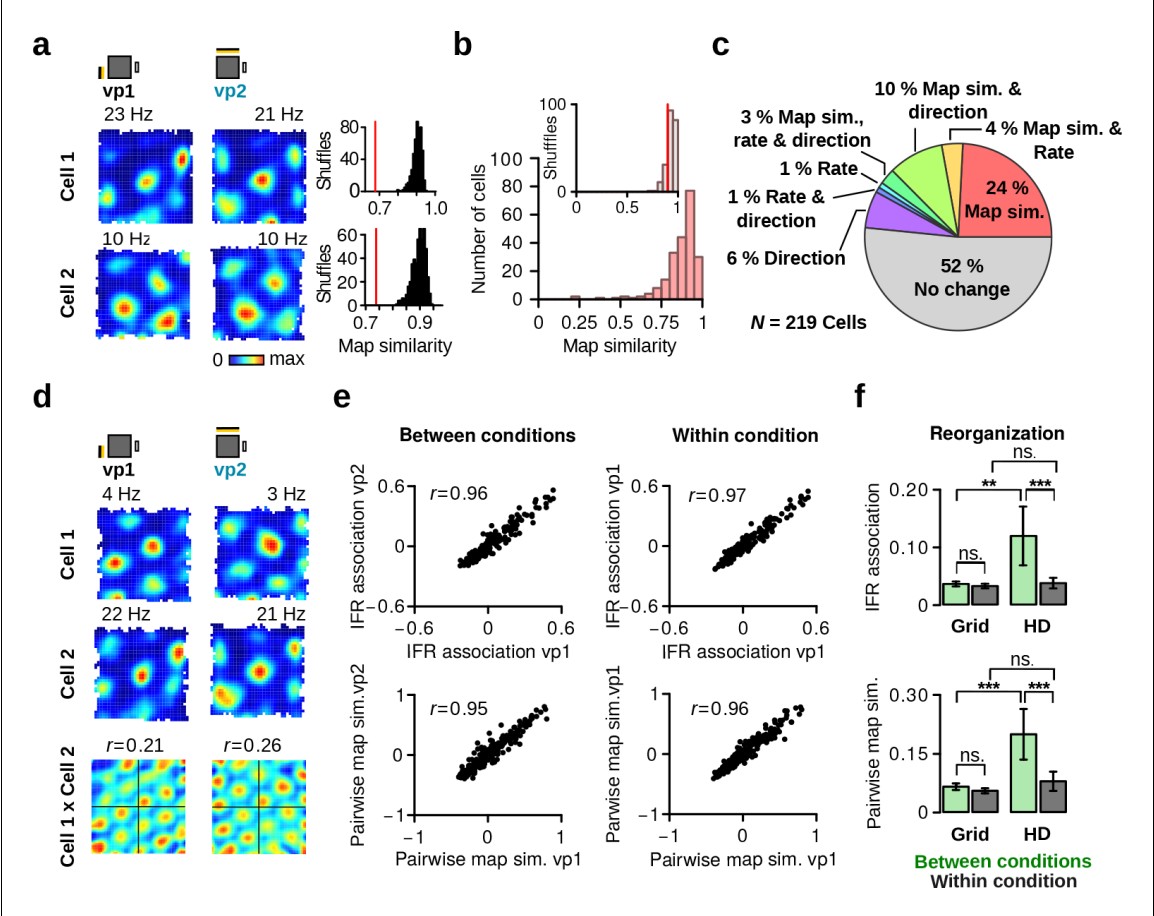

**Figure 8.** Grid cells retain stable firing associations. (**a**) Left: firing rate maps of two representative grid cells during vp1 (left) and vp2 (right) trials. Right: observed map similarity between vp1 and vp2 trials (red line) and distribution of map similarity when trial labels were reassigned randomly. (**b**) Distribution of map similarity for all grid cells (red). Inset: the median of observed map similarity (red line) with the distribution expected by chance (gray). (**c**) Pie chart illustrating the percentages of grid cells with a significant change in map similarity, average firing rate or preferred HD. (**d**) Firing rate maps of two simultaneously recorded grid cells and their spatial crosscorrelation. r values are correlation coefficients representing pairwise map similarity of two cells during each trial type. (**e**) Comparison of instantaneous firing rate (IFR) associations (top) and pairwise map similarity (bottom) of grid cell pairs during vp1 and vp2 trials (left, between conditions) or for two subsets of vp1 trials (right, within condition). (**f**) Reorganization of IFR associations (top) and pairwise map similarity (bottom) between conditions (vp1 vs. vp2) or within condition. Reorganization is shown separately for pairs of grid cells and pairs of HD cells. Plots show mean ±95% confidence intervals. ns.: not significant, **: p < 0.01, ***: p < 0.001.

DOI: https://doi.org/10.7554/eLife.35949.017

The following figure supplement is available for figure 8:

**Figure supplement 1.** Grid cell firing rate map examples and grid cell pairs with significant visually driven changes.

DOI: https://doi.org/10.7554/eLife.35949.018

the firing associations of non-rhythmic HD cells reorganized when visual landmarks were changed. This result indicates that the firing associations of non-rhythmic HD cells are malleable and controlled by visual landmarks.

A recent study also reported the existence of a population of HD cells in the dysgranular retrosplenial cortex that was controlled by local landmarks (*Jacob et al., 2017*). In a two-compartment environment in which the compartment cue cards had the opposite orientation to one another, some HD cells had unimodal tuning curves within each compartment but their preferred direction rotated by 180° when the animal moved across compartments. A second population of HD cells had bimodal tuning curves within each compartment, and their direction associated with the highest firing rate rotated 180° across compartments. Thus, the preferred direction of these cells was likely controlled by the visual landmarks in each compartment. Importantly, bidirectional cells were recorded simultaneously with classic HD cells that did not change their tuning curves across

compartments, demonstrating that bidirectional HD cells could re-align independently of the main HD system. The data currently available on non-rhythmic HD cells and bidirectional HD cells of the retrosplenial cortex suggest that, despite being both being controlled by visual landmarks, they might operate differently. One potential difference is their internal coherence as a population. Whereas bidirectional cells rotated coherently with one another when the animal moved across compartments, changing the visual landmarks in our experiment caused a reorganization of firing associations within the non-rhythmic HD cell population. Whether this difference originates from distinct cell properties will need to be investigated more directly by testing the two populations in similar recording conditions. Taken together, these findings suggest that the HD signal in cortical areas is more heterogeneous than in the anterodorsal thalamus or the presubiculum.

The different firing properties of theta-rhythmic and non-rhythmic HD cells suggest that they support different functions. Theta-rhythmic HD cells showed no reorganization during our experiment. This coherent HD signal could be well suited for providing directional information to grid cells. Indeed, computational models of grid cells assume that the HD signal remains coherent, and recordings from grid cells indicates that they do not show major reorganization after visual cue manipulations (*Hafting et al., 2005*; *Pérez-Escobar et al., 2016*). The theta modulation of rhythmic HD cells might also play an important role in the integration of the HD signal by grid cells. It has previously been proposed that the HD signal needs to be transformed into a theta-modulated signal to be used for navigation (*Cacucci et al., 2004*). Moreover, the segregation of HD cell ensembles during individual theta cycles might also contribute to grid periodicity (*Brandon et al., 2013*). We therefore suggest that an important function of theta-rhythmic HD cells is to provide a directional input to the grid cell network.

One function of non-rhythmic HD cells might be to act as an interface between visual landmarks and the coherent HD signal (*Bicanski and Burgess, 2016*). These non-rhythmic HD cells show large, non-coherent tuning changes upon landmark manipulations. An interesting mechanism that links visual landmarks to the HD signal has been put forward by Jacob and colleagues based on their recordings in the dysgranular retrosplenial cortex (*Jacob et al., 2017*; *Page and Jeffery, 2018*). In this model landmark-controlled HD cells could mediate a two-way interaction between visual landmarks and a coherent HD signal. This idea can be applied to the two HD cell populations identified in our study. The tuning curves of non-rhythmic HD cells would be established partly based on landmark-dependent inputs. The HD selectivity of these cells would make them co-active with a subgroup of classic HD cells (e.g. theta-rhythmic HD cells) with similar HD preferences. Over time, if the visual landmarks controlling the non-rhythmic HD cells remain stable, the association between the non-rhythmic HD cells and the coherent HD signal would be strengthened, allowing stable visual landmarks to gain preferential control over the coherent HD signal. This way, the population activity of landmark-controlled HD cells could contribute to setting the preferred directions of theta-rhythmic HD cells when an animal enters visually distinct environments. One interesting possibility is that this interaction between non-rhythmic and rhythmic HD cells might contribute to the orientation of grid cells within each grid module (*Stensola et al., 2012*).

Our data suggest that theta-rhythmic and non-rhythmic HD cells could form anatomically distinct populations. The differences in spike waveforms indicate that the two populations have distinct intrinsic biophysical properties. Moreover, the low incidence of pair recordings comprising a theta-rhythmic and a non-rhythmic cell HD suggests that the two populations are only partially overlapping in the MEC/PaS area. Histological analysis also points to the possibility that non-rhythmic cells could be preferentially located in the dorsal-most portion of the MEC. This could potentially correspond to the large cytochrome-oxidase positive, putatively parasubicular, patches previously described (*Burgalossi et al., 2011*). The high degree of HD selectivity in anatomically identified neurons located in these patches is consistent with this idea, although we note that most neurons in the large patches had strong theta modulation (*Burgalossi et al., 2011*).

In conclusion, our results establish that HD cells of the MEC/PaS comprise two functionally distinct types of cells. Theta-rhythmic HD cells react coherently when visual landmarks are altered and resemble more closely classic HD cells recorded in the anterodorsal thalamic nucleus or presubiculum. In contrast, non-rhythmic HD cells not only show larger responses to visual landmark manipulations, but also respond non-coherently to these manipulations. This study therefore reveals the existence of landmark-driven reorganization within a population of HD cells. Previous work has shown that non-periodic place selective neurons of the MEC react to alterations in the recording

environment by shifting the location of their firing fields (*Diehl et al., 2017*). The findings presented here indicate that an equivalent context-dependent reorganization occurs within the MEC/PaS HD code.

## Materials and methods

Data analysis was performed in the R software environment and the source code is available from GitHub (*Kornienko and Allen, 2018*; copy archived at https://github.com/elifesciences-publications/prog_kornienko_2018). The raw data (spike trains, position data and histology) of this study are available from the Dryad Digital Repository (doi:10.5061/dryad.202m6h0) (*Kornienko et al., 2018*).

### Subjects

The subjects were nine 3- to 6-month-old male wild-type C57BL/6 mice. They were singly housed in 26 cm x 20 cm x 14 cm high cages containing 2 cm of saw dust and one facial tissue. Mice were kept on a 12 hr light-dark schedule with all procedures performed during the light phase. All experiments were carried out in accordance with the European Committees Directive (86/609/EEC) and were approved by the Governmental Supervisory Panel on Animal Experiments of Baden Württemberg in Karlsruhe (35–9185.81/G-50/14).

### Surgical procedure

Mice were implanted with two 4-tetrode microdrives, targeting the MEC/PaS of each hemisphere. The microdrives allowed independent movement of individual tetrodes, which were made from four 12-$\mu$m tungsten wires (California Fine Wire Company). Mice were anaesthetized with isoflurane (1–3%) and fixed to the stereotaxic instrument. The skull was exposed and four miniature screws were inserted into the skull. Two screws located above the cerebellum served as ground electrodes. The skull above the MEC/PaS was removed and the tetrode bundles were implanted at the following coordinates (ML: ±3.1 mm from the midline, AP: 0.2 mm anterior from the transverse sinus, 6° angle in the posterior direction). Once the tetrode tips were 0.8 mm into the cortex, the microdrives were fixed to the skull with dental cement. During the first 72 hr post-surgery, mice received a s.c. injection of buprenorphine (0.1 mg/kg; Temgesic) every 8 hr. Mice were given a week to recover after surgery.

### Recording system, spike extraction and spike clustering

Mice were connected to the data acquisition system (RHD2000-Series Amplifier Evaluation System, Intan Technologies, analog bandwidth 0.09–7603.77 Hz, sampling rate 20 kHz) via a lightweight cable. Electrode signals were amplified and digitized by two 16-channel amplifier boards (RHD2132, Intan Technologies). Recording was controlled using ktan software (https://github.com/kevin-allen/ktan; copy archived at https://github.com/elifesciences-publications/ktan). Action potential detection was performed offline from the bandpass-filtered signal (800–5000 Hz). Waveform parameters were extracted with principal component analysis. Clusters of spikes were automatically generated using Klustakwik (https://sourceforge.net/projects/klustakwik/), before being manually refined with a graphical interface program.

Clusters quality was estimated from the spike-time autocorrelation and isolation distance. A refractory period ratio was calculated from the spike-time autocorrelation from 0 to 25 ms (bin size: 0.5). The mean number of spikes from 0 to 1.5 ms was divided by the maximum number of spikes in any bin between 5 and 25 ms. Any cluster with a refractory period ratio larger than 0.125 was discarded. Likewise, clusters with an isolation distance smaller than five were discarded.

The position and HD of the mouse was estimated from the position of two infrared-LEDs (wave length 940 nm), one large and one small, that were attached to one amplifier. The large and small LEDs were located anterior and posterior to the center of the head, respectively. The distance between the LEDs was approximately 8 cm. An infrared video camera (resolution of 10 pixels/cm, DMK 23FM021, The Imaging Source) located directly above the recording environment monitored the LEDs at 50 Hz. The location and HD of the mouse were extracted on-line from the position of the LEDs (https://github.com/kevin-allen/positrack; copy archived at https://github.com/elifesciences-publications/positrack).

## Initial training

Initial training began one week after surgery and took place in a different room than that used for recording sessions. Food intake was controlled to reduce the weight of the mice to 85% of their normal free-feeding weight. Mice were trained three times per day for 10 min to retrieve food rewards (AIN-76A Rodent tablets 5 mg, TestDiet) delivered at random locations within a 70 × 70 cm open field. After 2 days of training, the training time was extended (3 × 15 min) and the microdrives were connected to the recording system. This procedure continued until the mice explored the entire open field within 15 min. On each day, the raw signals were monitored on an oscilloscope and tetrodes were lowered until large theta oscillations were visible on most tetrodes.

## Recording environment

The recording environment consisted of an elevated gray square platform (70 × 70 cm) made of PVC. The platform was surrounded by four 40-cm-high walls located 10 cm away from the edges of the platform. A standard white cue card (21 × 29.7 cm) was attached to the center of one wall and remained in place throughout the experiment. Two visual patterns made of LED strips (color temperature: 3000 K, Ribbon Slim Top, Ledxon Group, powered by six 1.2 V batteries) were affixed to two other adjacent walls. Visual pattern 1 (vp1) consisted of 4 horizontal 20-cm-long rows (5 cm between rows) starting 7 cm away from the proximal junction of two walls. Visual pattern 2 (vp2) was a single horizontal 80-cm-long row of 48 LEDs. These two visual patterns were the only sources of visible light in the recording environment, and were switched on and off via a relay switch operated by a microcontroller (Arduino Uno). The four walls with the affixed visual patterns were stationary and had a constant orientation within the recording room.

The recording environment was surrounded by opaque black curtains. Food rewards (AIN-76A Rodent tablets 5 mg, TestDiet) were delivered at random locations from a pellet dispenser located above the ceiling of the recording environment (CT-ENV-203–5 pellet dispenser, MedAssociates). The pellet dispenser was controlled by a microcontroller (Arduino Uno) and the inter-delivery intervals ranged from 20 to 40 s.

## Recording protocol

The mouse was brought to the recording room at the beginning of the recording session and vp1 was turned on. The mouse was placed on the square platform and allowed to forage for 20 min. The presented visual pattern alternated between vp1 and vp2 every 2 min. There was no dark period between vp1 and vp2 trials because previous work showed that the spatial representation of grid cells and HD cells can be impaired after only few seconds in darkness (*Pérez-Escobar et al., 2016*; *Chen et al., 2016*). After 20 min, the mouse was taken off of the platform and placed in a small rest box (23 × 25 × 30 cm) for 20 min. No food reward was available in the rest box. The rest of the recording session comprised a series of 20 min blocks with the following sequence: foraging-rest-foraging-rest-foraging. In each recording session, the mouse foraged on the square platform for a total of 80 min. At the end of the recording session, the tetrodes were lowered by 25–50 $\mu$m.

## Instantaneous firing rate

The instantaneous firing rate (IFR) of a cell was obtained by counting the number of spikes in 1 ms time bins and applying a Gaussian smoothing kernel (standard deviation of 200 ms) to this spike vector. The firing probability was integrated over 100 ms time windows and transformed to a firing rate.

## Identification of spatially selective neurons

Most functional cell types were identified from their HD tuning curves or firing rate maps. A HD tuning curve consisted of the firing rate of a cell as a function of HD (10° bins). An occupancy vector containing the time in seconds spent within the different HD bins was calculated and smoothed with a Gaussian kernel (standard deviation of 10°). The number of spikes in each bin was then divided by the occupancy vector to obtain the firing rate for different HDs. The firing rate vectors were smoothed with a Gaussian kernel (standard deviation of 10°).

Firing rate maps were generated by dividing the square platform into 2 × 2 cm bins. The time in seconds spent in each bin was calculated and this occupancy map was smoothed with a Gaussian kernel (standard deviation of 3 cm). The number of spikes in each bin was divided by the smoothed

occupancy map to obtain a firing rate map. A smoothing kernel (standard deviation of 3 cm) was applied to the firing rate map. Only periods when the mouse ran faster than 3 cm/s were considered.

To identify HD cells, vp1 and vp2 tuning curves were calculated by concatenating data from all vp1 and all vp2 trials, respectively. A HD score was defined as the mean vector length of the tuning curve. The preferred direction of the neuron was the circular mean of the tuning curve. To be selected, HD cells had to have a HD score larger than 0.4 and a peak firing rate larger than 5 Hz during vp1 trials or vp2 trials or both. The arbitrary threshold of 0.4 is more conservative than most thresholds obtained with a shuffling procedure. The arbitrary threshold was used to ensure that neurons classified as HD cells had strong HD selectivity and clear preferred HDs.

A third criterion was added to the definition of HD cells to ensure that their HD tuning curve was not simply a consequence of spatial selectivity coupled with a biased sampling of HD on the recording platform. This was required because the behavior of mice is not completely random and not all HDs are equally sampled at every position on the platform. We therefore applied the distributive hypothesis method (*Muller et al., 1994*; *Cacucci et al., 2004*) which measured the similarity between the observed HD tuning curve and an expected tuning curve derived from the firing rate map of the cell and the HD occupancy probability at every bin of the firing rate map. The predicted HD tuning curve was defined as follows:

$$R_{Pred}(\theta) = \sum (R_P T_P(\theta)) / \sum T_P(\theta),$$

where $R_P$ was the firing rate in one bin of the firing rate map and $T_P(\theta)$ was the time spent facing HD $\theta$ in that bin. A distributive ratio, $DR$, was then calculated to estimate the similarity between the observed and predicted HD tuning curve:

$$DR = \sum |ln((1 + R_{Obs}(\theta))/(1 + R_{Pred}(\theta)))|/N,$$

where $N$ was the number of bins in a HD tuning curve. A DR of zero indicated that the observed HD tuning curve was well predicted by a combination of spatial selectivity and bias in HD sampling. A higher value indicated that the HD tuning curve was poorly predicted by the spatial selectivity of the cell and that its firing rate was modulated by HD. Cells had to have a DR larger than 0.2 to be considered HD cells.

Grid cells were identified based on the periodicity in each firing rate map. A spatial autocorrelation matrix was calculated from the firing rate map. Peaks in the autocorrelation matrix were defined as more than 10 adjacent bins with values larger than 0.1. The 60° periodicity in the spatial autocorrelation matrix was estimated using a circular region of the spatial autocorrelation matrix containing up to six peaks and excluding the central peak. Pearson correlation coefficients ($r$) were calculated between this circular region and a rotated version of the same region (by 30°, 60°, 90°, 120°, and 150°). A grid score was obtained from the formula:

$$\left(\frac{r_{60°} + r_{120°}}{2}\right) - \left(\frac{r_{30°} + r_{90°} + r_{150°}}{3}\right).$$

Significance thresholds for grid scores were obtained by shifting the position data by at least 20 s before recalculating grid scores. This procedure was repeated 100 times for each neuron to obtain surrogate distributions. The 99th percentiles of the null distributions were used as cell-specific significance thresholds.

Spatial selectivity was measured using a sparsity score, adapted so that high scores reflected high sparsity:

$$1 - \left(\frac{\left(\sum_{i=1}^{N} p_i \lambda_i\right)^2}{\sum_{i=1}^{N} p_i \lambda_i^2}\right)$$

where $N$ was the number of bins in the firing rate map, $p_i$ and $\lambda_i$ were the occupancy probability and firing rate in bin $i$, respectively. Significance levels for sparsity score were obtained with the same shuffling procedure as for grid scores.

The firing rate modulation by running speed was estimated using the speed score (*Kropff et al., 2015*), which was defined as the correlation coefficient between the running speed of the animal

and the IFR of a neuron. The running speed of the animal was obtained for these 100 ms time bins and a Pearson correlation was performed between the running speed and the IFR. Only time windows in which the running speed was between 3 and 100 cm/s were considered. Significance levels for speed scores were obtained with the same shuffling procedure as for the grid scores.

## Theta rhythmicity and theta oscillations

The theta rhythmicity of neurons was estimated from the instantaneous firing rate of the cell. The number of spikes observed in 1 ms time window was calculated and convolved with a Gaussian kernel (standard deviation of 5 ms). The firing probability was integrated over 2 ms windows and transformed into a firing rate. A power spectrum of the instantaneous firing rate was calculated using the *pwelch* function of the *oce* R package. The estimates of the spectral density were scaled by multiplying them by the corresponding frequencies: $spec(x)*freq(x)$. A theta rhythmicity index for each neuron was defined as $\frac{\theta-baseline}{\theta+baseline}$, where $\theta$ is the mean power at 6–10 Hz and *baseline* is the mean power in two adjacent frequency bands (3–5 and 11–13 Hz). The theta rhythmicity indices of HD cells were analyzed with the *Mclust* function of the R package *mclust* which uses Gaussian mixture modeling and the EM algorithm to estimate the number of components in the data.

Theta oscillations were detected using one wire per tetrode. The signal was bandpass filtered at delta (2–4 Hz) and theta (6–10 Hz) frequencies and the power of the filtered signals (root mean square) was calculated in non-overlapping 500 ms windows. Windows with a theta/delta power ratio larger than two were defined as theta epochs. Individual theta cycles within theta epochs were identified using the bandpass filtered (5–14 Hz) signal. The positive-to-negative zero-crossing in the filtered signal delimited individual theta cycles and were assigned phases 0 and 360. The theta phase of spikes was linearly interpolated within the cycle boundaries. For each neuron, the modulation of firing rate by theta oscillations was quantified by calculating the mean vector length of the theta phases assigned to the spikes. The circular mean of the spikes served as the preferred theta phase of a neuron.

## Theta-cycle skipping

A theta-cycle skipping index was calculated from the spike-time autocorrelation (2 ms bin, 0–300 ms) of each cell. The peaks $p1$ between 100–150 ms and $p2$ between 200–300 were identified and the index was defined as: $(p2-p1)/max(p1,p2)$. Only cells with a theta-cycle skipping index above 0.1 were considered theta-cycle skipping cells (*Brandon et al., 2013*).

## Statistical significance of visually driven changes

To determine if a HD cell significantly changed its preferred HD, HD score, or mean firing rate between vp1 and vp2 trials, a shuffling procedure was used. The trial labels vp1 and vp2 were reassigned randomly to the forty 2 min trials before recalculating vp1 and vp2 tuning curves. The differences in preferred HD, HD score and mean firing rate between the reassigned vp1 and vp2 trials were calculated. These last steps were repeated 500 times to obtain a distribution of changes expected to occur by chance for the three variables. The observed changes of the HD cell were considered significant if they were larger than 99% of the differences obtained with the shuffling procedure. Because the firing properties of individual cells influenced the shuffled distributions, the changes observed in one cell were only compared to the shuffled data of that same cell.

## Egocentric landmark bearing tuning curves

Egocentric landmark bearing tuning curves were calculated to assess the firing rate of a cell as a function of the angular position of a landmark in the mouse's field of view. Visual pattern 1 (vp1) was used as landmark (*Figure 5—figure supplement 1*). For each time point $t$, we obtained the angle $\phi$ between the position of the animal $x_0, y_0$ and the landmark position at $x_1, y_1$. After calculating $dx = x_1 - x_0$, $dy = y_1 - y_0$, $\phi$ was defined as the angle between the animal's location and $d_x, d_y$. To obtain the egocentric landmark bearing angle $\alpha$ representing the angular location of the landmark in the animal's field of view, the observed head direction angle at time $t$ was subtracted from $\phi$. Egocentric landmark bearing tuning curves were calculated the same way as allocentric HD tuning curves using $\alpha$ instead of the observed head direction.

## Bidirectionality of HD tuning curves

Bidirectionality in HD tuning curves was determined from a bidirectionality (BD) score. The BD score was defined as the firing rate at the second largest peak of the HD tuning curve divided by the firing rate at the largest peak of the tuning curve $BD\,score = min(p1, p2)/max(p1, p2)$. To be considered a peak, the firing rate had to be above 2 Hz. Bidirectional HD cells were identified as HD cells for which the BD score was larger than 0.2 and the ratio between the firing rate of the second largest peak and the trough in the HD tuning curve with the highest firing rate exceeded 1.25.

## Firing association and reorganization of cell pairs between trial types

The IFR association of two neurons was estimated by the correlation coefficient between their respective IFR vectors. Reorganization scores were defined to investigate whether the tuning curve changes in pairs of cells were coherent.

For a cell pair, cell $i$ and $j$, the preferred HD (*PrefHD*) reorganization score between conditions was defined as follows:

$$|(PrefHD_{j,vp1} - PrefHD_{i,vp1}) - (PrefHD_{j,vp2} - PrefHD_{i,vp2})|$$

As a control, a reorganization score was calculated using two mutually exclusive subsets of vp1 trials ($vp1.1$ and $vp1.2$):

$$|(PrefHD_{j,vp1.1} - PrefHD_{i,vp1.1}) - (PrefHD_{j,vp1.2} - PrefHD_{i,vp1.2})|$$

A similar procedure was used to calculate reorganization of HD scores by replacing the preferred HD by HD score. Reorganization of IFR associations was defined as the absolute difference between IFR associations observed during vp1 and vp2 trials.

## Theta synchrony

Co-modulation of spike activity by theta oscillations for pairs of HD cells was estimated from their spike-time crosscorrelations (−300 to 300 ms, bin size: 2 ms). Each crosscorrelation was transformed into a vector of Z-scores. A discrete wavelet transform (*WaveletComp* R package) was applied to the Z-score vector. Theta synchrony was defined as the mean power in the time widow of 0±25 ms at theta frequency (6 to 10 Hz).

## Histology

At the end of the experiment, mice were deeply anesthetized with an i.p. injection of ketamine/xylazine, and perfused with saline followed by 4% paraformaldehyde. The brains were removed and stored at 4°C in 4% paraformaldehyde for at least 24 hr before being sectioned in 50-$\mu$m-thick slices and stained with cresyl violet. All brain sections were digitized with a motorized widefield slide scanner (Axio Scan.Z1, Zeiss).

# Acknowledgements

We thank H Monyer, MI Schlesiger, and DAA MacLaren for their constructive comments on the manuscript, and JA Pérez-Escobar, J Ivanova and Y Wang for their technical assistance. This work was supported by an Emmy Noether Program grant (AL 1730/1-1) and a Collaborative Research Centre (SFB-1134) from the Deutsche Forschungsgemeinschaft.

# Additional information

## Funding

| Funder | Grant reference number | Author |
| --- | --- | --- |
| Deutsche Forschungsgemeinschaft | AL 1730/1-1 | Kevin Allen |
| Deutsche Forschungsgemeinschaft | SFB1134 | Kevin Allen |

The funders had no role in study design, data collection and interpretation, or the decision to submit the work for publication.

## Author contributions
Olga Kornienko, Conceptualization, Software, Formal analysis, Visualization, Methodology, Writing—original draft, Writing—review and editing; Patrick Latuske, Laura Kohler, Investigation, Writing—review and editing; Mathis Bassler, Investigation; Kevin Allen, Conceptualization, Software, Formal analysis, Supervision, Funding acquisition, Methodology, Writing—original draft, Writing—review and editing

## Author ORCIDs
Olga Kornienko http://orcid.org/0000-0003-1936-8414
Kevin Allen http://orcid.org/0000-0001-5319-3721

## Ethics
Animal experimentation: All experiments were carried out in accordance with the European Committees Directive (86/609/EEC) and were approved by the Governmental Supervisory Panel on Animal Experiments of Baden Wurttemberg in Karlsruhe (35-9185.81/G-50/14).

## Decision letter and Author response
Decision letter https://doi.org/10.7554/eLife.35949.025
Author response https://doi.org/10.7554/eLife.35949.026

## Additional files

### Supplementary files
• Supplementary file 1. Histological results. Location of the tetrode tips and number of HD cells (theta-rhythmic and non-rhythmic) and grid cells recorded in each hemisphere.
DOI: https://doi.org/10.7554/eLife.35949.019

• Transparent reporting form
DOI: https://doi.org/10.7554/eLife.35949.020

### Data availability
All data generated or analysed during this study are available.

The following dataset was generated:

| Author(s) | Year | Dataset title | Dataset URL | Database, license, and accessibility information |
| --- | --- | --- | --- | --- |
| Kornienko O, Latuske P, Bassler M, Kohler L, Allen K | 2018 | Data from: Non-rhythmic head-direction cells in the parahippocampal region are not constrained by attractor network dynamics | http://dx.doi.org/10.5061/dryad.202m6h0 | Available at Dryad Digital Repository under a CC0 Public Domain Dedication |

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
