## [Decision Letter]

Thank you for submitting your article "Head-direction cells escaping attractor dynamics in the parahippocampal region" for consideration by *eLife*. Your article has been reviewed by three peer reviewers, and the evaluation has been overseen by a Reviewing Editor and Michael Frank as the Senior Editor. The following individuals involved in review of your submission have agreed to reveal their identity: Adrien Peyrache (Reviewer #1); Shane M O'Mara (Reviewer #2); Kate J Jeffery (Reviewer #3).

The reviewers have discussed the reviews with one another and the Reviewing Editor has drafted this decision to help you prepare a revised submission.

The referees all found the results, showing differential responses to a change in visual landmark configuration, which would not be possible within the simplest interpretation of the ring attractor model, quite interesting. Moreover, the mapping onto sub populations of theta and non-theta modulated head direction cells is of significant interest. However, they also recommend some revisions aimed at clarification and increasing the framing with respect to previous findings. Thus, that fact that head direction cell firing can be modulated by factors external to the attractor network itself needs to be raised earlier, most obviously the previous experimental work by Jacob et al., 2017 which showed similar differential responses of sub-populations of head direction cells to visual landmarks. Indeed, it might be worth pointing out that any attractor model of head-direction cell firing must include inputs from cells encoding the angles to specific landmarks for the spatial stability of the attractor tuning. Some other factors to address include whether the directional tuning might vary with uncertainty, e.g. depending on the size of the angle to the landmark, or to the width of the landmark. Please details see below.

Essential revisions:

1) Much of the Introduction and Discussion focuses on the independence of the landmark-sensitive directional neurons from the putative head direction attractor network. This is interesting, but it is not novel, since it was reported by Jacob et al. for the RSC head direction cells (indeed that paper is entitled "An independent, landmark-dominated head-direction signal…".) The present cells are highly reminiscent of those cells and seem to have similar properties – this fact is mentioned only briefly and yet is surely relevant and important, since RSC is one of the big inputs to MEC/PaS. The closeness of the properties of these MEC/PaS neurons to the dysgranular RSC ones is sufficiently striking as to be meaningful, and the theta modulation differences have a potentially important message about the way in which these subsystems interact with the hippocampal formation.

2) The truly novel observation, which is that the landmark-preferring neurons are not theta-modulated while the "normal" HD cells are, suggesting they are possibly in different circuits, is relegated to second place and is not even mentioned in the title.

The current presentation of manuscript is hard to follow. First, the authors report that HD cells changed their properties depending on the landmarks. Then, in Figure 6, when the cells are divided between theta and non-theta modulated, the difference seems to be of another kind: in the theta group, cells seem pretty stable, and in the non-theta group, some show changes larger than the theta group (which could then be considered as the 'chance level'). Presenting Figure 5 with a color code indicating theta and non-theta pairs would help. But it would make the paper easier to follow by presenting from the beginning the distinction between the two classes of neurons.

3) There is also a third observation, that some cells switch on and off in response to the landmarks, that is barely mentioned, and yet is novel and important and additionally separates these neurons from "ordinary" HD cells. The switching on and off of some cells suggests that – as the authors themselves note – the cells are “driven” by landmarks rather than merely anchored by them.

The Abstract makes much reference to "HD cells". However given the findings then presented, one starts to wonder whether "head direction cells" is not the right term – that they can follow visual landmarks around so easily suggests they are more interested in the visual scene than head direction per se. And yet, without the landmark manipulations, these certainly look like classic HD cells and have been labelled so previously. I wonder, though, about calling them something slightly more vague – directional cells perhaps – and then, at the end, suggesting a new name – landmark directional cells?

Overall, the bi- or multi- directionality of the cells in the presence of multiple landmark configurations requires some greater theoretical consideration. The obvious point to be made regards the need for all attractor systems for stable spatial coding to be anchored to environmental cues, which requires an interaction with cells coding for the directions of landmarks, and is not a trivial problem given the parallax problem (e.g. Bicanski and Burgess, 2016)

4) Can the authors present evidence that theta and non-theta neurons are not intrinsically different? (e.g. waveforms, firing rates, and putative layers / distance from dorsal border, if available?). There is a basic level of phenotyping required – and the authors need to discuss possibilities here a little more than they have.

5) Overall, the number of cells (and pairs) is rather small. Can the authors demonstrate that, at least, the off-diagonal outliers in Figure 5B (which are likely responsible for the strong decrease in the reported correlation values) are from more than one or two neurons, and from different animals if possible?

We need to be told (Results section) how many cells were recorded in each animal.

6) Another possible interpretation for differences in tuning is that animals are more 'lost' in one condition than in the other. One way to control for this would be to check whether the change in tuning depends on the preferred direction of the neurons. For example, a HD cell pointing in the opposition direction of the white card in the vp2 condition may be more affected than a cell pointing to the white card just because, in average, more error accumulates in this direction. The authors should report that the preferred direction of the HD neurons that showed changes between the two conditions is homogeneously distributed.

Along the same lines, the work of Acharya et al., 2016 suggests that, for HD cells of the hippocampus, the width of a visible landmark has an influence on the cell's tuning (i.e. HD score) and direction (toward the landmark). This paper, by the way, was already a first demonstration that, in some conditions, certain HD neurons are not rigidly coordinated. If one assumes that the LED arrays are more salient than the white card, it is then possible that the effect of vp1 or vp2 on the HD network is different. Although the two examples shown in Figure 2B suggest that instead the change can occur in both ways (lower or higher HD score), was it the case at the population level? Have the authors observed any redistribution of preferred direction toward the smaller landmark (vp1)?

Why no manipulations of ambient lighting? One particular concern is that the LED strips were switched from vp1 to vp2 in the presence of the animal – with no intervening period of darkness. It is possible that there will be important differences in firing activity depending on prior visual circumstance – there are substantial visual inputs to the areas recorded from. This should be discussed.

7) Title – this needs a revision. As stated it can be read, not unreasonably, as an unmasking phenomenon resulting from the presence of an inhibitor in a particular projection. Something along the lines of 'Theta-rhythmic and non-theta-rhythmic HD cells in PHC do not show attractor dynamics”.

---

## [Author Response]

Essential revisions:1) Much of the Introduction and Discussion focuses on the independence of the landmark-sensitive directional neurons from the putative head direction attractor network. This is interesting, but it is not novel, since it was reported by Jacob et al. for the RSC head direction cells (indeed that paper is entitled "An independent, landmark-dominated head-direction signal…") The present cells are highly reminiscent of those cells and seem to have similar properties – this fact is mentioned only briefly and yet is surely relevant and important, since RSC is one of the big inputs to MEC/PaS. The closeness of the properties of these MEC/PaS neurons to the dysgranular RSC ones is sufficiently striking as to be meaningful, and the theta modulation differences have a potentially important message about the way in which these subsystems interact with the hippocampal formation.

We thank the reviewers for this important point and agree that the work of Jacob et al., 2017 was somewhat overlooked. The Introduction has now been considerably modified to better frame our work with respect to previous findings. The work of Jacob et al., 2017 is discussed at length, with a particular focus on the idea that attractor networks of HD cells require interactions with visual landmark information in order to remain stable (Bicanski and Burgess, 2016). Also, we now put less emphasis on the independence of the non-rhythmic HD cells from a putative HD attractor network.

As pointed out by the reviewers, the two principal similarities between bidirectional (BD) cells (Jacob et al., 2017) and non-rhythmic HD cells is that visual landmarks control their preferred direction and that their activity is not coherent with the main HD system. However, we note one potentially important difference in the internal dynamics of the two populations. In the work of Jacob et al., 2017, between-compartment BD cells showed a 180^o^ rotation in their tuning curves when the animal moved from one compartment to the other. Similarly, most of the within-compartment BD cells showed a 180^o^ rotation of their dominant peak between compartments (their Figure 3C). Thus, as a population, BD cells appeared to rotate coherently. In contrast, the non-rhythmic HD cell population showed non-coherent changes and non-rhythmic HD cells changed their firing associations with one another. We have now added a paragraph in the Discussion to highlight the similarities and differences between BD cells and non-rhythmic HD cells. Of course, a clear assessment of the differences between BD cells and non-rhythmic HD cells will ultimately require recording the two cell populations in the same experimental conditions.

2) The truly novel observation, which is that the landmark-preferring neurons are not theta-modulated while the "normal" HD cells are, suggesting they are possibly in different circuits, is relegated to second place and is not even mentioned in the title.

The presentation of the important theta-rhythmic / non-rhythmic dissociation now comes much earlier in the Results section. In addition, the title now reads “Non-rhythmic head-direction cells in the parahippocampal region are not constrained by attractor network dynamics”.

The current presentation of manuscript is hard to follow. First, the authors report that HD cells changed their properties depending on the landmarks. Then, in Figure 6, when the cells are divided between theta and non-theta modulated, the difference seems to be of another kind: in the theta group, cells seem pretty stable, and in the non-theta group, some show changes larger than the theta group (which could then be considered as the 'chance level'). Presenting Figure 5 with a color code indicating theta and non-theta pairs would help. But it would make the paper easier to follow by presenting from the beginning the distinction between the two classes of neurons.

As suggested by the reviewers, we have modified the order of presentation in the Results section.

3) There is also a third observation, that some cells switch on and off in response to the landmarks, that is barely mentioned, and yet is novel and important and additionally separates these neurons from "ordinary" HD cells. The switching on and off of some cells suggests that – as the authors themselves note – the cells are “driven” by landmarks rather than merely anchored by them.

We now mention that we observed HD cells that almost turned silent with one light pattern at the end of the Introduction, in the Results section, and in the Discussion.

The Abstract makes much reference to "HD cells". However given the findings then presented, one starts to wonder whether "head direction cells" is not the right term – that they can follow visual landmarks around so easily suggests they are more interested in the visual scene than head direction per se. And yet, without the landmark manipulations, these certainly look like classic HD cells and have been labelled so previously. I wonder, though, about calling them something slightly more vague – directional cells perhaps – and then, at the end, suggesting a new name – landmark directional cells?

We have considered at length the possibility of giving non-rhythmic HD cells a different name reflecting their potential function. We did not want to use the term “directional cells” as this is very vague and the analysis is all based on the direction of the head. We are also hesitant to call them “landmark directional cells” as it suggests that the cells encode egocentric landmark bearing. We now show in the revised manuscript that this is unlikely to be the case (see next point below). For this reason, we decided to keep the term “non-rhythmic HD cells” until we have a more complete understanding of their function. If the reviewers think that another term would be more appropriate, we can, of course, modify the manuscript accordingly.

Overall, the bi- or multi- directionality of the cells in the presence of multiple landmark configurations requires some greater theoretical consideration. The obvious point to be made regards the need for all attractor systems for stable spatial coding to be anchored to environmental cues, which requires an interaction with cells coding for the directions of landmarks, and is not a trivial problem given the parallax problem (e.g. Bicanski and Burgess, 2016).

We have now modified the Abstract and Introduction to highlight the importance of visual landmarks in anchoring an attractor network of HD cells to the external world.

In addition, we investigated the possibility that non-rhythmic HD cells were in fact egocentric landmark bearing cells, as described in Bicanski and Burgess, 2016. Egocentric landmark bearing cells fire when a landmark is located at a given position in the animal’s field of view.

We set 8 possible anchor points (putative landmarks) located 8 cm away from the edge of the arena (Figure 5—figure supplement 1). We calculated an egocentric landmark bearing tuning curve for each of these points. The best egocentric landmark bearing tuning score (vector length) out of the 8 scores served as egocentric score. In theory, if an egocentric landmark-bearing cell has a sharp place field, its best egocentric tuning curve resembles its allocentric HD tuning curve (Bicanski and Burgess, 2016). However, if an egocentric landmark bearing cell fires throughout an environment, its egocentric tuning curve becomes narrower than its allocentric HD tuning curve. This is because the position of a visual landmark in the animal’s field of view when an animal faces in a specific direction will vary with the position of the animal.

We therefore tested whether the improvement from allocentric HD score to egocentric score was larger for non-rhythmic HD cells with low spatial information scores. This is not what we observed. The improvement from allocentric HD score to egocentric score was not significantly related to the information score of the HD cell (Figure 5—figure supplement 1C, r = -0.004, P = 0.98). Therefore, it seems unlikely that most non-rhythmic HD cells were egocentric landmark bearing cells.

4) Can the authors present evidence that theta and non-theta neurons are not intrinsically different? (e.g. waveforms, firing rates, and putative layers / distance from dorsal border, if available?). There is a basic level of phenotyping required – and the authors need to discuss possibilities here a little more than they have.

We now provide a more thorough description of the two HD cell classes. We added new figure panels (Figure 3 D-F) to further characterize the intrinsic properties of the two HD cell populations and refer to these findings in the Discussion section.

The average firing rates of theta-rhythmic and non-rhythmic HD cells were not significantly different (Wilcoxon signed-rank test, N = 34 and 59, w = 993, P = 0.94). This is now shown in Figure 3C, right panel.

We investigated whether the two cell populations could be partially anatomically segregated within the MEC/PaS. By analyzing pairs of HD cells simultaneously recorded on the same tetrode (N = 17), we found only one mixed pair. This suggests that the two cell populations are at least partially anatomically segregated. In addition, we found that fewer non-rhythmic HD cells were recorded on same tetrodes as grid cells compared to theta-rhythmic HD cells (N = 9 and 42, respectively) (Chi-squared test: Χ²= 15.66, df = 1, P < 10^-5^). Finally, non-rhythmic HD cells tended to be recorded during earlier recording sessions compared to theta-rhythmic HD cells (Figure 3D, w = 497, P < 10^-5^), suggesting that non-rhythmic HD cells are found more dorsally and/or in deeper layers.

We decided to implant additional mice (N = 4 mice) to obtain clearer histological data regarding the anatomical localization of non-rhythmic HD cells. The minimal spacing between tetrodes was increased to approximately 0.50 mm, making it possible to assign each cell to a tetrode track. These mice were recorded on a similar protocol and the experiments stopped as soon as non-rhythmic HD cells were encountered and the tetrodes were not moved further. Although this approach has a sampling bias (i.e., only the most dorsal non-rhythmic cells will be recorded), it provides a finer localisation of the recorded cells. In these mice, we found that non-rhythmic HD cells are found in the dorsal part of the MEC (Figure 3E and Figure 3—figure supplement 2A). Interestingly, we also observed non-rhythmic HD cells in the postrhinal cortex (two mice).

Taken together, the results suggest that non-rhythmic HD cells are mainly located near the dorsal border of the MEC/PaS. In contrast, theta-rhythmic HD cells appeared more equally spread along the dorso-ventral axis of the MEC/PaS.

Finally, analysis of the spike waveforms in the two populations revealed that non-rhythmic HD cells had longer spike durations (Figure 3F, N = 30 and 52, w = 1347, P < 10^-8^) and larger spike asymmetry compared to theta-rhythmic HD cells (Figure 3F, w = 463, P = 0.002). Cells which had inverted waveforms (N = 4 non-rhythmic and 7 theta-rhythmic) were excluded from this analysis.

5) Overall, the number of cells (and pairs) is rather small. Can the authors demonstrate that, at least, the off-diagonal outliers in Figure 5B (which are likely responsible for the strong decrease in the reported correlation values) are from more than one or two neurons, and from different animals if possible?

The effect presented in Figure 5B (now Figure 7B) was not caused by one or two neurons, or by the data from only one mouse. We have re-plotted the same data and used a different color for each mouse (Author response image 1). The plots show that the data points further away from the diagonal in Figure 5B (now Figure 7B) are from more than one animal.

**Author response image 1. respfig1:** Correlation between the differences in preferred HD, in HD score and in instantaneous firing rate (IFR) association of HD cell pairs during vp1 and vp2 trials. A unique color is used for each mouse. Solid red lines are regression lines and dashed red lines represent confidence intervals.

We need to be told (Results section) how many cells were recorded in each animal.

This information is provided in Supplementary File 1 (see “All cells” column). We have added a sentence that points to Supplementary File 1 in the Results section.

6) Another possible interpretation for differences in tuning is that animals are more 'lost' in one condition than in the other. One way to control for this would be to check whether the change in tuning depends on the preferred direction of the neurons. For example, a HD cell pointing in the opposition direction of the white card in the vp2 condition may be more affected than a cell pointing to the white card just because, in average, more error accumulates in this direction. The authors should report that the preferred direction of the HD neurons that showed changes between the two conditions is homogeneously distributed.

To address this point, we first tested whether the mice might be more lost in one condition than in the other. This analysis is presented in Figure 4—figure supplement 2. We first compared the HD score of HD cells between vp1 and vp2 trials for cells with a significant tuning curve change and found no significant differences (Figure 4—figure supplement 2A; paired Wilcoxon signed-rank test for HD score, N = 59, v = 995, P = 0.41). We then reported the preferred direction of the HD neurons that showed changes between the two conditions (Figure 4—figure supplement 2B). The preferred directions of the neurons were widely distributed. A test of uniformity revealed that the preferred directions deviated marginally from uniformity during vp1 trials but not during vp2 trials (Rayleigh test of uniformity, vp1: test statistic = 0.2303, P = 0.0437, mean resultant length = 0.23, vp2: test statistic = 0.1116, P = 0.4798, mean resultant length = 0.11). There was no clear accumulation of preferred directions at directions associated with vp1, vp2 or the cue.

Along the same lines, the work of Acharya et al., 2016 suggests that, for HD cells of the hippocampus, the width of a visible landmark has an influence on the cell's tuning (i.e. HD score) and direction (toward the landmark). This paper, by the way, was already a first demonstration that, in some conditions, certain HD neurons are not rigidly coordinated. If one assumes that the LED arrays are more salient than the white card, it is then possible that the effect of vp1 or vp2 on the HD network is different. Although the two examples shown in Figure 2B suggest that instead the change can occur in both ways (lower or higher HD score), was it the case at the population level? Have the authors observed any redistribution of preferred direction toward the smaller landmark (vp1)?

We thank the reviewers for pointing out the work of Acharya et al., 2016. This study is highly relevant to our work and is now cited in the Results section.

As suggested by the examples shown in Figure 4B, both increase and decrease in HD score could be observed from vp1 to vp2 trials. This resulted in similar HD scores for vp1 and vp2 trials for those HD cells with significant tuning curve alterations (Figure 4—figure supplement 2A) or all HD cells (Figure 1—figure supplement 2F).

There was no clear accumulation of the preferred direction for directions associated with vp1 and vp2 (Figure 4—figure supplement 2B). When considering all HD cells, the distribution of preferred direction was homogeneous in both trial types (Rayleigh test of uniformity, vp1: test statistic = 0.0931, P = 0.4466; vp2: test statistic = 0.0479, P = 0.81). This finding suggests no accumulation of preferred directions towards the smaller landmark. The HD scores as a function of preferred direction for HD cells changing their tuning curves are shown in Figure 4—figure supplement 2C. A linear-circular correlation suggests that the two variables are only weakly correlated (vp1: R = 0.07 P = 0.02; vp2: R = 0.09, P = 0.005). There was no clear trend for sharper HD curves towards or away from the visual patterns.

Furthermore, the change in HD tuning was only weakly correlated with the preferred direction of the neurons for vp1 trials (Figure 4—figure supplement 2D, vp1: R = 0.06, P = 0.03). Lastly, these two variables were not correlated during vp2 trials (vp2: R = 0.035, P = 0.13).

Why no manipulations of ambient lighting? One particular concern is that the LED strips were switched from vp1 to vp2 in the presence of the animal – with no intervening period of darkness. It is possible that there will be important differences in firing activity depending on prior visual circumstance – there are substantial visual inputs to the areas recorded from. This should be discussed.

We agree with the reviewers that the sequence of light presentation and the absence of dark trials are important factors (Taube et al., 1990, Knierim et al., 1995; Jeffery et al., 1999; Jeffery et al., 2016). Our aim was that the spatial representation of grid cells remained anchored to the static cues (i.e., cue card, uncontrolled cues on the floor) when the visual patterns switched. We assumed that this would be more likely if the animal had a precise estimate of its position when the light switch occurred. In previous work, we and others found that after a few seconds in darkness, the spatial representation of grid cells in mice is severely impaired (Pérez-Escobar et al., 2016, Chen et al., 2016). Therefore, we decided to leave the dark period out of the current recording protocol. We have added a sentence in the Materials and methods section to justify this choice in our paradigm (subsection “Recording protocol”).

In cue rotation experiments, the rotated and stationary cues compete for their influence on spatially selective cells (Taube et al., 1990, Knierim et al., 1995; Jeffery et al., 1999; Jeffery et al., 2016). For example, when a cue is rotated in the presence of the animal, the preferred direction of HD cells often under-rotate (Taube et al., 1990), indicating that the control of the cue on the HD cells is reduced. A similar effect has been observed for place cells (Jeffery et al., 1999). One interpretation of these results is that when a cue is perceived as unstable, either by being directly seen moving or by being in conflict with an internal path integrator, it has less influence on the HD or place cell activity. In our experiment, the two distinct landmarks had stable spatial locations and, as long as the mouse could discriminate one from one another, provided reliable spatial information when turned on. Whether the influence of spatial representation would be more pronounced if darkness periods were to be introduced between the vp1 and vp2 trials is something that remains to be tested.

7) Title – this needs a revision. As stated it can be read, not unreasonably, as an unmasking phenomenon resulting from the presence of an inhibitor in a particular projection. Something along the lines of 'Theta-rhythmic and non-theta-rhythmic HD cells in PHC do not show attractor dynamics”.

The title was changed to “Non-rhythmic head-direction cells in the parahippocampal region are not constrained by attractor network dynamics”.